# The EYA3 tyrosine phosphatase activity promotes pulmonary vascular remodeling in pulmonary arterial hypertension

Yuhua Wang[1], Ram Naresh Pandey[1], Allen J. York[2], Jaya Mallela [1], William C. Nichols[3], Yueh-Chiang Hu [1], Jeffery D. Molkentin [2], Kathryn A. Wikenheiser-Brokamp[4] & Rashmi S. Hegde [1]

In pulmonary hypertension vascular remodeling leads to narrowing of distal pulmonary arterioles and increased pulmonary vascular resistance. Vascular remodeling is promoted by the survival and proliferation of pulmonary arterial vascular cells in a DNA-damaging, hostile microenvironment. Here we report that levels of Eyes Absent 3 (EYA3) are elevated in pulmonary arterial smooth muscle cells from patients with pulmonary arterial hypertension and that EYA3 tyrosine phosphatase activity promotes the survival of these cells under DNA-damaging conditions. Transgenic mice harboring an inactivating mutation in the EYA3 tyrosine phosphatase domain are significantly protected from vascular remodeling. Pharmacological inhibition of the EYA3 tyrosine phosphatase activity substantially reverses vascular remodeling in a rat model of angio-obliterative pulmonary hypertension. Together these observations establish EYA3 as a disease-modifying target whose function in the pathophysiology of pulmonary arterial hypertension can be targeted by available inhibitors.

[1] Division of Developmental Biology, Cincinnati Children's Hospital Medical Center, Department of Pediatrics, University of Cincinnati College of Medicine, 3333 Burnet Avenue, Cincinnati, OH 45229, USA. [2] Heart Institute, Cincinnati Children's Hospital Medical Center, University of Cincinnati College of Medicine, 3333 Burnet Avenue, Cincinnati, OH 45229, USA. [3] Division of Human Genetics, Cincinnati Children's Hospital Medical Center, Department of Pediatrics, University of Cincinnati College of Medicine, 3333 Burnet Avenue, Cincinnati, OH 45229, USA. [4] Division of Pathology & Laboratory Medicine and Perinatal Institute, Division of Pulmonary Biology, Cincinnati Children's Hospital Medical Center, Department of Pathology and Laboratory Medicine, University of Cincinnati College of Medicine, 3333 Burnet Avenue, Cincinnati, OH 45229, USA. Correspondence and requests for materials should be addressed to R.S.H. (email: rashmi.hegde@cchmc.org)

Pulmonary Hypertension (PH), regardless of cause, is characterized by changes in the structure of the pulmonary vasculature (pulmonary vascular remodeling) and inappropriate vasoconstriction. Treatment of PH typically begins with primary therapy directed toward the underlying cause. However, there are no effective primary therapies for most types of Pulmonary Arterial Hypertension (PAH, WHO group 1 PH). PAH treatment is directed towards the symptoms and largely consists of vasodilators accompanied by supportive regimens. With disease progression, vasoconstriction becomes less important and vascular remodeling becomes the predominant contributor to the pathology. Reversal of remodeling and hence restoration of vascular structure and function is the unmet therapeutic goal[1]. Here we report that inhibition of the Eyes Absent (EYA3) tyrosine phosphatase can attenuate DNA-damage repair, promote apoptosis of pulmonary vascular cells, and substantially reverse established vascular remodeling in a rodent model of experimental pulmonary hypertension.

DNA damage[2–6], altered DNA-damage control[3,7], and chronic inflammation[8–10] are present in PAH lungs. Despite these challenges pulmonary vascular cells resist apoptosis and proliferate. Such sustained proliferation and escape from apoptosis in the face of DNA-damaging events are among the hallmarks of cancer cells, and PAH has often been described as a cancer-like disease that could be targeted by anti-neoplastic agents[11–14]. To counter DNA damage, both cancer and pulmonary vascular cells trigger a DNA-damage response that senses and repairs DNA lesions. Inhibiting DNA-damage repair represents a therapeutic opportunity in both diseases.

A central player in the repair of DNA double-stranded breaks is the minor histone protein H2AX. H2AX is constitutively phosphorylated on Tyrosine-142 ($Y_{142}$)[15] and dephosphorylation of Y142 is required for the assembly of functional DNA-damage repair complexes and cell survival[16,17]. The EYA family of proteins are tyrosine phosphatases (PTPs)[18,19] that dephosphorylate H2AX at Y142 permitting the assembly of DNA-damage repair (DDR) foci nucleated by γ-H2AX (Ser139 phosphorylated form of H2AX)[16]. There are four EYA family members. Of these EYA1, 2 & 4 play developmental roles[19], but are not present in most normal adult tissue. The PTP activity of the EYAs is mechanistically unique: a nucleophilic Aspartate residue initiates a two-step, divalent metal-dependent dephosphorylation reaction[18]. This unusual mechanism makes it possible to specifically target the EYA PTPs without inhibiting the over 120 classical PTPs that use a Cysteine-based reaction mechanism and share a common active-site stereo-chemistry. Elevated levels of EYA proteins in cancer cells are linked to increased tumor growth[20–24], metastasis[25], and resistance to DNA damaging therapeutics[26], and endothelial EYA3 promotes tumor angiogenesis[24]. Hence the EYA proteins are being actively investigated as targets for the development of cancer therapeutics.

Here we show that EYA3 is present in pulmonary vascular cells, and that EYA3 levels are elevated in pulmonary arterial smooth muscle cells (PASMC) from patients with PAH. Furthermore, inhibition of the EYA-PTP activity with Benzarone (BZ; a previously characterized EYA-PTP inhibitor[24,27–29]) can both attenuate the development of experimental pulmonary hypertension and substantially reverse established pulmonary hypertension in a rodent model. Reduced levels of γ-H2AX foci, decreased proliferation and increased apoptosis are seen in lung tissue upon treatment with BZ. Hence BZ directly targets the hyper-proliferative, apoptosis-resistant patho-phenotype. The increase in pulmonary arterial pressure after the Sugen-hypoxia (Su-Hx) protocol is also significantly lower in mice with a single amino-acid substitution in EYA3 that inactivates the tyrosine phosphatase activity, providing genetic evidence that EYA3-PTP

activity contributes to the development of experimental PH. Together these observations support the EYA3 PTP as a molecular target for the development of therapeutics directed toward vascular remodeling, the defining pathology of PAH.

## Results

**EYA3 levels and DNA damage in PASMC.** PASMC from idiopathic PAH patients (labeled L10, L85) were obtained from the Pulmonary Hypertension Breakthrough Initiative, phenotype was confirmed by positive staining for α-smooth muscle actin, and cells were used between passages 3 and 8 for all studies (Supplementary Table 1, Supplementary Fig. 1). Normal human PASMC were purchased as controls. RT-PCR showed that normal adult human PASMC had Eya3 transcript but no transcript for Eya1, Eya2, or Eya4 (Supplementary Fig. 1g). PAH-PASMC had higher levels of EYA3 protein relative to normal PASMC as quantified on western blots (representative gel in Fig. 1a). Pulmonary arterial endothelial cells from idiopathic PAH patients (L89, L105) and normal controls (PAEC) were similarly examined. Only Eya3 transcript was detected in PAEC (Supplementary Fig. 1h), and similar levels of EYA3 protein were detected in normal and PAH-PAEC (Fig. 1b). To determine the clinical relevance of elevated EYA3 levels in PAH-PASMC we examined lung tissue from PH patients for EYA3 expression around vascular lesions. Serial sections were stained for the smooth muscle cell marker α-smooth muscle actin (α-SMA; Fig. 1c, e, g) and EYA3 (Fig. 1d, f, h). While EYA3 is present in multiple cell types, the levels of EYA3 are significantly higher in α-SMA-positive cells surrounding small pulmonary arterioles in PH lungs compared with control lungs (representative images from a healthy control (Fig. 1c, d) and two representative PH patients (Fig. 1e–h) are shown).

We next examined constitutive DNA damage in PAH-PASMC cells by two methods: the alkaline COMET assay which is a direct and sensitive measure of DNA damage and staining for γ-H2AX DDR foci which reports on the consequences of double-strand breaks (detection and repair). Patient-derived PASMC had higher percentages of cells with γ-H2AX foci (Fig. 1i, k) and increased DNA damage as measured by COMET tail moments (Fig. 1j, l). These results are consistent with a previous report of elevated γ-H2AX foci in PASMC from PAH patients[7].

**EYA3 promotes PASMC survival in the presence of DNA damage.** The question of how PAH-PASMC adapt to DNA damage is a subject of extensive investigations[3,7,11,30]. Here we asked whether the ability of PAH-PASMC to survive under DNA damaging conditions correlates with the presence of EYA3 PTP activity. We used the prototypical oxidant $H_2O_2$ (200 μM $H_2O_2$ for 60 min) to induce DNA damage as evidenced by a strong γ-H2AX signal (Supplementary Fig. 2j). Cell survival was then monitored by the use of the WST-8 reagent. Only 50% of normal PASMC survived $H_2O_2$ treatment (Fig. 2a). In contrast, exposure to $H_2O_2$ had a much less significant effect on PAH-PASMC; over 75% PAH-PASMC remained viable after $H_2O_2$ treatment (Fig. 2a). To determine whether this effect correlates with EYA3 levels we established L10 and L85 cell lines expressing two different shRNA targeting Eya3 or a non-specific shRNA control. Changes in EYA3 protein levels were confirmed by western blot (Supplementary Fig. 2a, b). Loss of EYA3 resulted in a significantly reduced ability to survive after $H_2O_2$ treatment (Fig. 2c, d and Supplementary Fig. 2f, g). To test the hypothesis that the EYA PTP activity might contribute to the ability of PAH-PASMC to survive DNA damage we used the previously characterized small molecule inhibitor benzarone (BZ)[24,27–29]. In the presence of BZ, PAH-PASMC survival after exposure to $H_2O_2$ was reduced

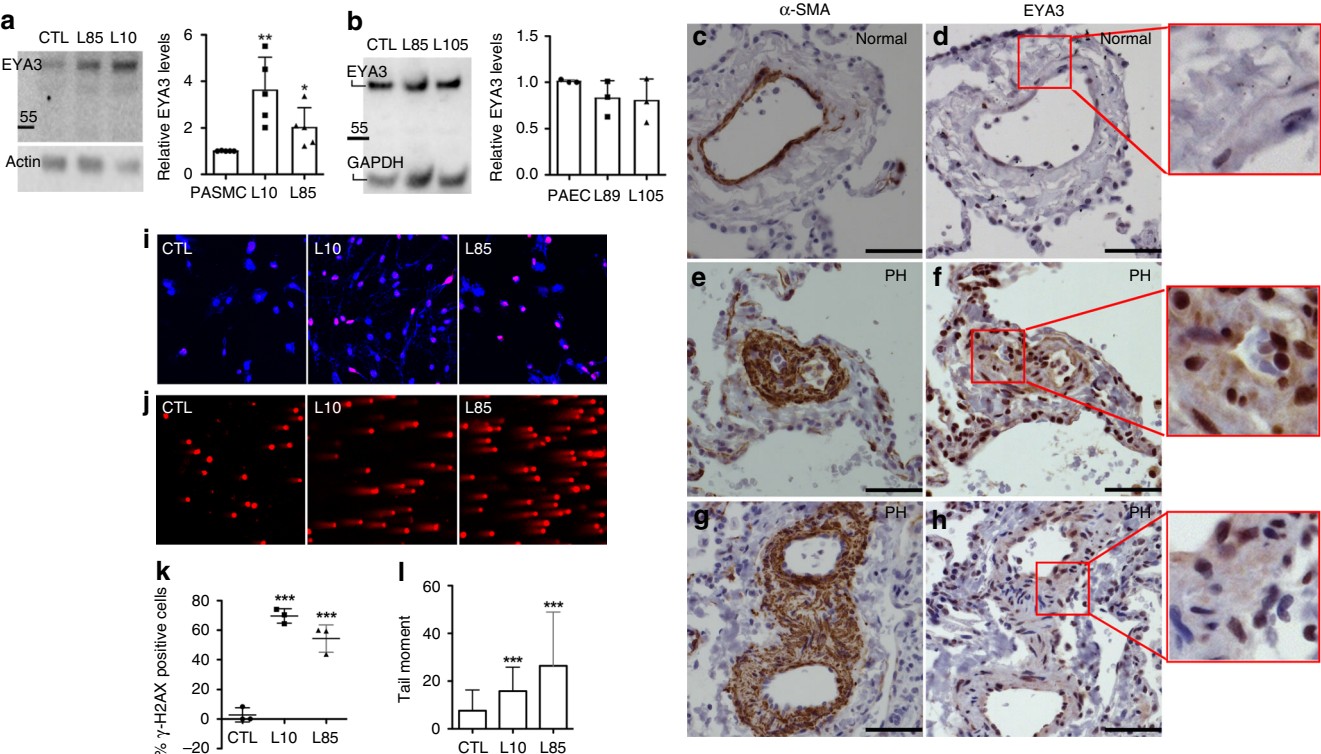

**Fig. 1** PAH-PASMC have elevated levels of EYA3 and DNA damage. **a**, **b** Western blot analyses of primary human PASMC and PAEC from normal lungs (CTL) and from the lungs of PAH patients (PAH-PASMC: L10, L85, PAH-PAEC: L89, L105). Blots were probed with anti-EYA3 antibody and quantitated using ImageJ, relative levels of EYA3 in multiple experiments (PASMC $n = 5$, PAEC $n = 3$) are shown in the bar graphs (plotted as mean ± SD; statistical significance (normal versus PAH-PASMC) was determined using two-tailed $t$-tests, **$P < 0.01$, *$P < 0.05$). EYA3 levels were consistently higher in PAH-PASMC compared with normal PASMC. In contrast, EYA3 levels in PAH-PAEC were not elevated relative to normal PAEC. **c**–**h** EYA3 protein levels are elevated in plexiform lesions of lungs from PH patients. Representative serial immunohistochemical sections stained with antibodies toward α-SMA (**c**, **e**, **g**) and EYA3 (**d**, **f**, **h**). Human lung tissue explants were obtained from the Cincinnati Children's Hospital Medical Center Transplant Service; normal human lung tissue (transplant donor, 62-year-old female; **c**, **d**) and two patients with PH (**e**, **f** 14-year-old female; **g**, **h** 8-year-old female) are shown. Scale bars = 50 μm. Higher magnification of the areas enclosed by red boxes in (**d**), (**f**), (**h**) shown to the right. (**i**)PAH-PASMC have higher levels of γ-H2AX than normal (CTL) subjects. Cells were stained with anti-γ−H2AX antibody (red) and DAPI (blue). **j** Alkaline COMET assays show increased tail moments in PAH-PASMC. **k** Percentage of γ-H2AX foci-positive cells is elevated in PAH-PASMC compared with normal PASMC (CTL). The number of cells containing >10 γ-H2AX foci/nucleus were counted using ImageJ. The data represent the average of independent experiments. Graph shows mean ± SD. One-way ANOVA with Dunnett's post-test was conducted. ***$P < 0.001$. **l** Tail moments in PAH-PASMC compared with normal PASMC (CTL). Tail moments calculated using OPENCOMET on >100 cells per sample and two independent experiments. Graph shows the median with range. One-way ANOVA with Dunnett's post-test was conducted. ***$P < 0.001$. Source Data are provided as a Source Data file

to levels comparable to that of normal PASMC (Fig. 2b). BZ treatment of either L10-shEYA3 or L85-shEYA3 did not further reduce cell survival (Fig. 2e, f and Supplementary Fig. 2h, i), suggesting that the effect of BZ in these experiments is EYA3-dependent. These observations support a role for the EYA PTP activity in permitting survival of PASMC under DNA-damaging conditions. Interestingly, there was no difference in the survival of normal and PAH-PAEC in similar experiments, and BZ did not have any detectable effect on PAEC survival (Supplementary Fig. 2c–e).

**EYA-PTP inhibition reduces RVSP and RV hypertrophy.** In order to determine the effect of the EYA-PTP inhibitor BZ in vivo we used an experimental animal model of angio-obliterative pulmonary hypertension in rats, the Sugen-Hypoxia (Su-Hx) protocol[31] in which a single administration of the VEGF inhibitor SU-5416 is followed by 3 weeks of hypoxia (10% $O_2$) and PH develops between 3 and 5 weeks after rats are returned to room air[32]. To test the effect of EYA-PTP inhibition on the development of PH, BZ was administered every 3 days for 5 weeks after return to room air (prevention protocol Fig. 3a). The rats were

maintained for a further 3 weeks in room air before right ventricular systolic pressure (RVSP) was measured by right heart catheterization. To determine whether EYA-PTP inhibitor administration was effective when administered after PH is typically established (treatment protocol Fig. 3a), BZ administration was initiated 5 weeks after return to room air.

Serial lung sections stained with α-SMA and EYA3 antibodies showed that, as in the human tissue and PAH-PASMC (Fig. 1), levels of EYA3 were elevated in pulmonary vascular cells after Su-Hx (Fig. 3b). The mean RVSP in vehicle-treated Su-Hx rats was 39.4 mm Hg higher than in animals maintained in room air. RVSP elevation was attenuated in Su-Hx rats treated with BZ in both the prevention and treatment protocols (Fig. 3c). Right ventricular (RV) hypertrophy as measured by the Fulton's index (right ventricle/left ventricle plus septum weight) showed a similar trend; a mean increase of 20.8 in vehicle-treated Su-Hx animals relative to room air controls, while there was a much smaller increase in BZ-treated animals (8.6 in the prevention protocol, 6.6 in the treatment protocol; Fig. 3d).

With the dosing regimen used in this study there was no measurable liver toxicity. Serum Alanine Transaminase (ALT) activity was comparable in vehicle-treated and BZ-treated

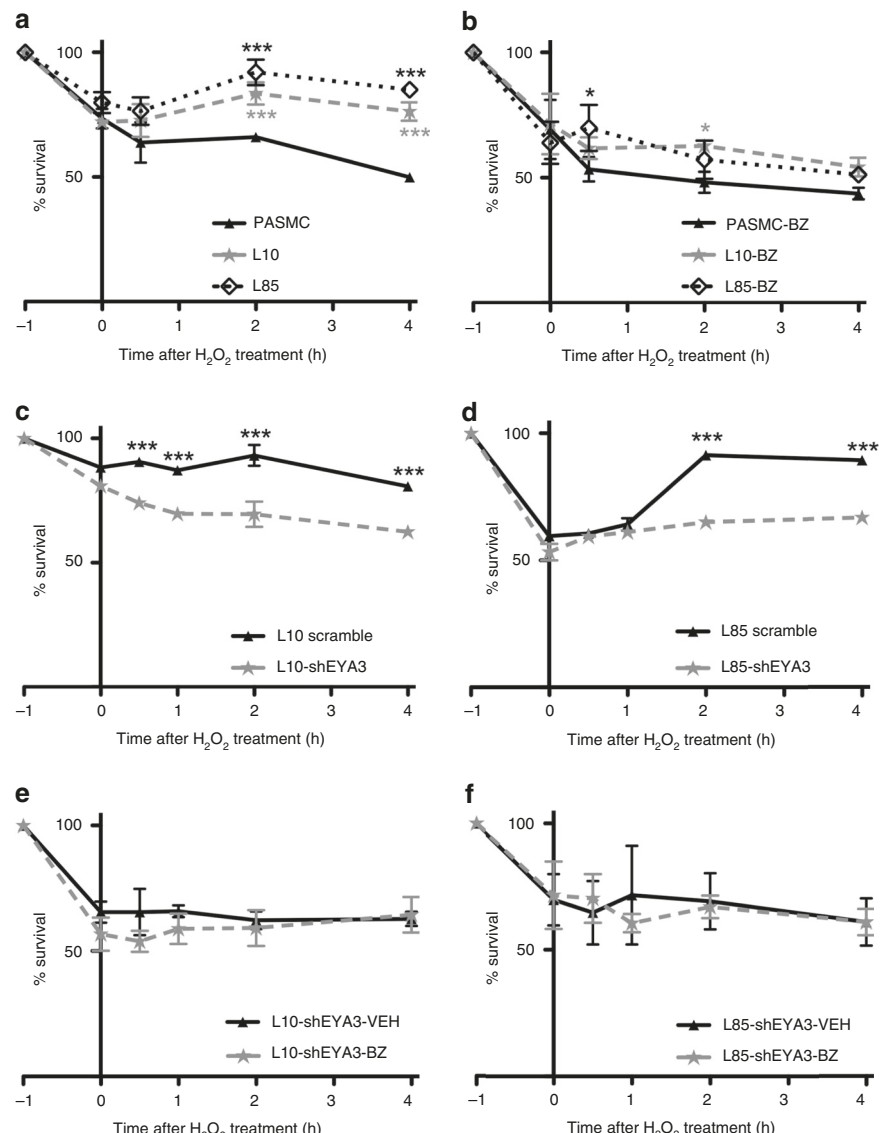

**Fig. 2** EYA3 PTP promotes survival of PASMC after DNA damage. In each experiment cells were treated with 200 μM $H_2O_2$ for 1 h. $H_2O_2$ was then withdrawn and the cells allowed to recover in normal culture medium. The percentage of viable cells (relative to untreated controls) were monitored using the WST-8 cell viability assay and are plotted versus time (x-axis). −1 h indicates the start of $H_2O_2$ treatment. $H_2O_2$ was withdrawn at 0 h. Each experiment was conducted at least three times. Representative data shown as the mean ± SD; statistical significance was determined using two-way ANOVA and Bonferroni's post-test, ***$P < 0.001$, *$P < 0.05$. **a** Recovery from $H_2O_2$ treatment of control PASMC and PAH-PASMC (L10 and L85) shows that PAH-PASMC rapidly recover relative to normal PASMC. **b** EYA-PTP inhibition with Benzarone makes PAH-PASMC susceptible to $H_2O_2$-induced DNA damage, while having no effect on control PASMC. **c** Stable knockdown of EYA3 in PAH-PASMC L10 cells increases their susceptibility to $H_2O_2$ treatment. **d** Stable knockdown of EYA3 in PAH-PASMC L85 cells increases their susceptibility to $H_2O_2$ treatment. **e** EYA-PTP inhibition with 7.5 μM BZ has no effect on the susceptibility of PAH-PASMC L10-shEYA3 cells to $H_2O_2$ treatment. **f** EYA-PTP inhibition with 7.5 μM BZ has no effect on the susceptibility of PAH-PASMC L85-shEYA3 cells to $H_2O_2$ treatment. Source Data are provided as a Source Data file

animals (Supplementary Fig. 3a). Furthermore, blinded analyses of liver sections conducted by a pathologist confirmed the lack of treatment-induced liver hepatoxicity. Livers from both treated and control animals had no, or minimal, histopathologic abnormalities consisting of only rare small foci of minimal lobular or portal tract inflammation and no necrosis, fibrosis, or neoplasia to indicate toxic injury. Cleaved caspase-3 staining showed no evidence of apoptosis in the liver (Supplementary Fig. 3b).

**EYA-PTP inhibition reduces vascular remodeling.** Pulmonary arterial lesions seen after Su-Hx had varying degrees of hyper-cellularity involving the media and adventitia of the vessels with some extension into the adjacent alveolar septa (Figs. 4a, 5b, g). Muscularized vessels and medial wall thickness were visualized using hematoxylin and eosin (H&E; Fig. 4a) and Elastica van Gieson (Fig. 4b) staining. Over 60% of blood vessels with diameter < 100 μm were muscularized in vehicle-treated Su-Hx rat lungs. BZ treatment reduced this to under 30% (Fig. 4c). The Su-Hx protocol resulted in a 4.5-fold increase in medial thickness of distal PAs while upon BZ treatment there was only between 1.5-fold (prevention protocol) and 1.7-fold (treatment protocol) increase (Fig. 4d). Co-staining of lung sections with the smooth muscle cell marker SM22-α and the endothelial cell marker von

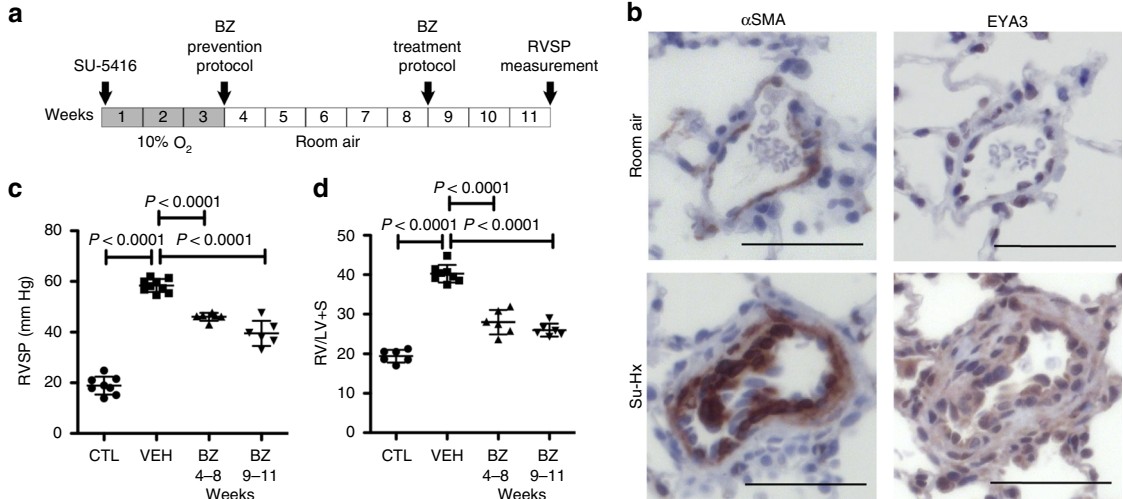

**Fig. 3** EYA-PTP inhibitor Benzarone reduces RVSP and RV hypertrophy. **a** Protocol for the rat Sugen-hypoxia (Su-Hx) model and BZ administration schedule. Adult rats were administered SU-5416, then placed in a hypoxia chamber for 3 weeks. After return to room air, start of BZ treatment (BZ or vehicle every 3 days for 5 weeks) for the "prevention" and "treatment" protocols is indicated by arrows. Eight weeks after return to room air right ventricular systolic pressure (RVSP) and right ventricular hypertrophy was assessed for all animals, including those maintained at room air as controls. There was no significant difference in either body weight (Su-Hx Veh 335.3 g ± 9.4, Su-Hx BZ 337.2 g ± 8.1) or heart weight (Su-Hx Veh 1.26 g ± 0.046, Su-Hx BZ 1.213 g ±0.026) between vehicle-treated and BZ-treated animals. **b** EYA3 levels in pulmonary vascular cells are elevated after the Su-Hx procedure. Representative serial sections from lung of rats maintained in room air or subject to the Su-Hx procedure. Sections were stained with antibodies toward α-SMA and EYA3. Scale bars = 50 μm. **c** BZ reduces right ventricular systolic pressure (RVSP). RVSP measured in rats maintained in room air (CTL), subject to the Su-Hx protocol and treated with vehicle upon return to room air (VEH), and treated with BZ in either the prevention or treatment protocol described in (**a**). Data plotted as the mean ± SD. Statistical significance was determined using $t$-tests to compare pairs of observations. CTL $n = 8$, VEH $n = 9$, BZ prevention $n = 6$, BZ treatment $n = 6$. **d** BZ reduces right ventricular hypertrophy. Right ventricular hypertrophy (Fulton's index = right ventricular (RV) weight/left ventricle (LV) + interventricular septum weight) measured 8 weeks after return to room air in rats maintained in room air (CTL), treated with vehicle starting upon return to room air after Su-Hx (VEH), and treated with BZ in either the prevention or treatment protocol described in (**a**). Data plotted as the mean ± SD. CTL $n = 6$, VEH $n = 8$, BZ prevention $n = 6$, BZ treatment $n = 6$. Source Data are provided as a Source Data file

Willebrand factor (vWF) confirmed muscularization of distal arterioles in vehicle-treated Su-Hx lungs and reduced muscularization with a relatively intact EC layer upon BZ treatment (Figs. 4e, 5a). Masson's trichrome staining of sections from the right ventricles showed significantly more fibrosis in Su-Hx vehicle-treated rats relative to those treated with BZ (Fig. 4f).

Approximately 10% of the cells (both α-SMA-positive and α-SMA-negative) in lungs subject to the Su-Hx protocol stained positive for Ki-67. In contrast, only rare proliferating cells were identified by Ki-67 staining in age-matched rats maintained in room air (Fig. 5c, d). BZ treatment reduced the overall proliferation in lungs of Su-Hx animals to background levels (Fig. 5c, d). Increased apoptosis was detected by cleaved caspase-3 (CC3) in sections from lungs of Su-Hx animals treated with BZ (Fig. 5e, f). Likely reflecting the time point of this analysis most of the CC3-positive cells were non-SMC.

Together these data provide evidence that EYA-PTP activity supports the survival and proliferation of pulmonary vascular cells and promotes vascular remodeling in the Su-Hx model.

**Loss of EYA3-PTP activity attenuates hypoxia-induced PH.** We next used a genetic strategy to evaluate the role of the EYA3-PTP activity in the pathogenesis of PH. CRISPR/Cas9 technology was used to generate C57BL/6 mice with a single amino-acid replacement in mouse EYA3 (Asp262 to Asn) that is known to abrogate tyrosine phosphatase activity[18]. The resultant mice, referred to as $Eya3^{D262N}$, have normal lung morphology and are viable and fertile. The C57BL/6 J background was used since these mice are known to develop moderate PH under hypoxia while other strains are less responsive[33]. To determine whether the EYA3 PTP activity contributes to the development of

experimental PH we used the mouse Su-Hx protocol; hypoxia for 3 weeks with weekly injections of anti-VEGF agent SU-5416. RVSP was measured as an indicator of pulmonary arterial pressure immediately upon return to room air after hypoxia. This timing is different from the rat Su-Hx protocol since PH resolves when mice are maintained in room air after hypoxia treatment[32]. Age-matched C57BL/6J ($n = 21$) and $Eya3^{D262N}$ ($n = 17$) mice were used in two independent experiments. Both male (11 C57BL6/J, 8 $Eya3^{D262N}$) and female (10 C57BL6/J, 9 $Eya3^{D262N}$) mice were included. There was no significant difference in either RVSP or Fulton's index between C57BL6/J and $Eya3^{D262N}$ animals maintained in room air. After the Su-Hx protocol the RVSP in control C57BL/6J mice was 21 mm Hg higher than for control animals maintained in room air (Fig. 6a), and the ratio of right ventricle to left ventricle plus septum weight [RV/(LV + Sep)] increased over 50%. In contrast, $Eya3^{D262N}$ mice exhibited only a 10.6 mm Hg increase in RVSP and <25% increase in RV hypertrophy (Fig. 6b). No significant differences in RVSP or Fulton's index were noted between male and female mice in this study.

Muscularization of small pulmonary arterioles was detected by immunoreactivity for α-SMA. Increased muscularization was present in $Eya3^{+/+}$ animals subject to the Su-Hx protocol but not in $Eya3^{D262N}$ animals (Fig. 6d, e). The proliferation index (percentage of Ki-67 positive cells) was increased in lung tissue after the Su-Hx protocol in control $Eya3^{+/+}$ mice, but remained unchanged in $Eya3^{D262N}$ mice (Fig. 6c, d). The status of the endothelial cell layer was analyzed by staining for vWF (Fig. 6e). While VEGF receptor blockade by SU-5416 induces endothelial cell apoptosis, the emergence and proliferation of an apoptosis-resistant sub-population of endothelial cells has been reported under conditions of chronic hypoxia[31,34]. At the time point evaluated in this study (upon completion of 3 weeks of hypoxia)

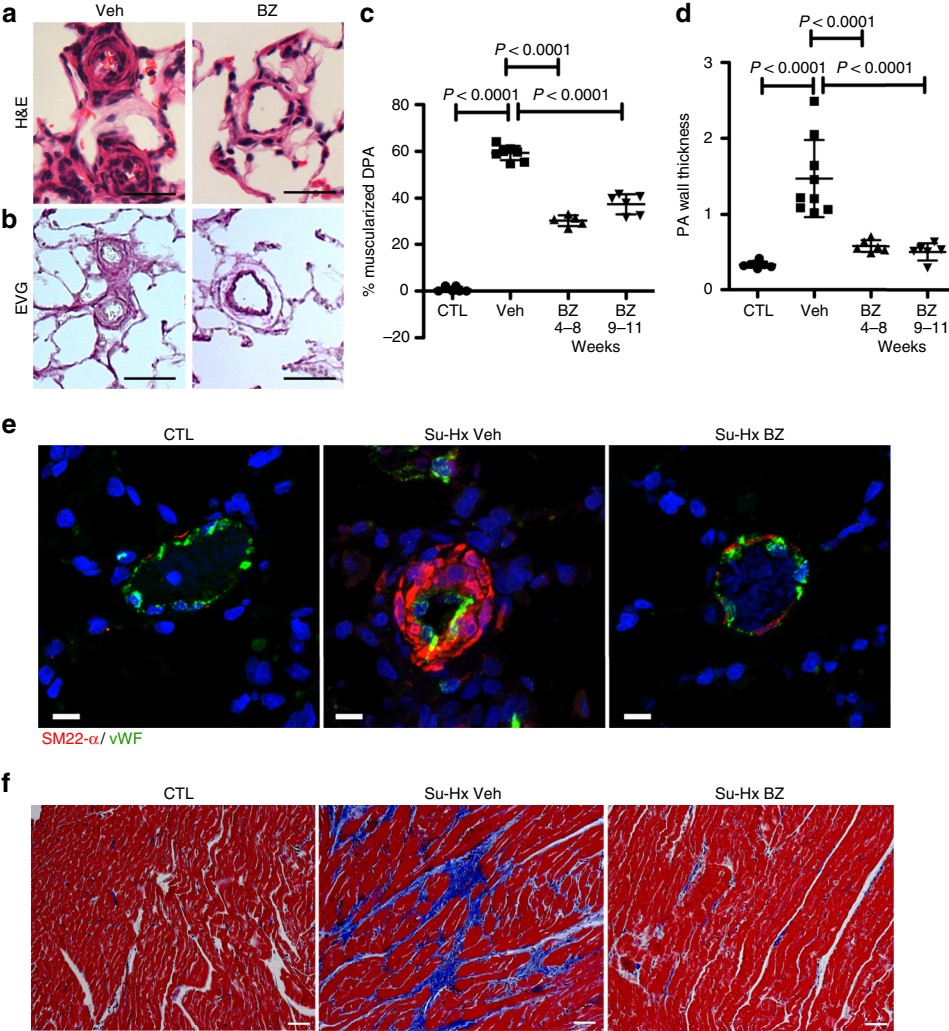

**Fig. 4** Benzarone treatment reduces muscularization of distal pulmonary arteries (DPA) in the Sugen-hypoxia (Su-Hx) protocol. **a** Representative histology of DPAs from vehicle-treated (Veh) and BZ-treated rat lungs subject to the Su-Hx protocol. Hematoxylin and eosin (H&E) stained sections are shown. Scale bar = 100 μm. **b** Representative histology of DPAs from vehicle-treated (Veh) and BZ-treated rat lungs subject to the Su-Hx protocol. Elastica van Gieson (EVG) stained sections are shown. Scale bar = 100 μm. **c** Percentage muscularized DPAs (diameter of 100 μm or below) in H&E stained rat lung sections are plotted. Each data point represents three sections per animal. CTL rats maintained in room air $n = 6$, VEH vehicle-treated animals subject to the Su-Hx protocol $n = 8$, BZ prevention–prevention protocol $n = 5$, BZ-treatment protocol $n = 6$. Data plotted as the mean ± SD, significance assessed by pairwise $t$-tests. **d** Pulmonary artery (PA) wall thickness in rat lung sections: difference in diameters defined by the lamina elastic externa and the lumen in two perpendicular directions of transversally cut vessels below 200 μm in diameter. Each data point represents an average of at least two sections per animal. CTL rats maintained in room air $n = 8$, VEH vehicle-treated animals subject to the Su-Hx protocol $n = 9$, BZ-prevention protocol $n = 6$, BZ treatment protocol $n = 6$. Data plotted as the mean ± SD, significance assessed by pairwise $t$-tests. **e** Representative sections from CTL (room air), Su-Hx rat lungs treated with vehicle (Su-Hx Veh), and Su-Hx lungs treated with BZ (treatment protocol; Su-Hx BZ) were stained for the smooth muscle cell marker SM22-α (red), the endothelial cell marker von Willebrand factor (vWF; green), and the nuclear marker DAPI (blue). Scale bar = 10 μm. **f** Representative images of Masson's Trichrome stained heart sections show fibrosis (blue) in the RV of Su-Hx–Veh rats that is significantly reduced upon BZ treatment. Scale bar = 100 μm. Source Data are provided as a Source Data file

the endothelial cell layer in both $Eya3^{+/+}$ and $Eya3^{D262N}$ mice subject to the Su-Hx protocol were similar to control mice maintained in room air (Fig. 6e) and cleaved caspase-3 staining did not detect any apoptosis.

Mouse models of PH have limitations; while increased RVSP has been reported in numerous studies immediately after 3 weeks in hypoxia, there is no evidence of significant or sustained angio-obliterative disease in mice[32,35]. Likewise, we did not observe either angio-obliteration or occluded lumens (Supplementary Figs. 4 and 5). Despite this limitation, the Su-Hx mouse model allows the use of transgenic mice and are an informative adjunct to the pharmacological approach used in the rat model.

**Attenuated DNA-damage repair upon loss of EYA3-PTP activity.** Two markers were used to evaluate DNA damage and repair (DDR) in lung tissue: γ-H2AX foci that report on the existence of DSBs, and 53BP1 foci that report on the assembly of functional DDR complexes. Rats subject to the Su-Hx protocol had increased γ-H2AX foci (Fig. 7a, c) and 53BP1 staining (Fig. 7d, f) around vascular lesions, likely as a consequence of ongoing DNA damage and repair. Rats treated with the EYA-PTP inhibitor BZ had an overall reduction in cells with γ-H2AX foci (Fig. 7a, c) and 53BP1 staining (Fig. 7d, f). Some residual γ-H2AX and 53BP1 staining was present in endothelial cells after BZ treatment (Fig. 7a, d) possibly indicating ongoing DNA damage

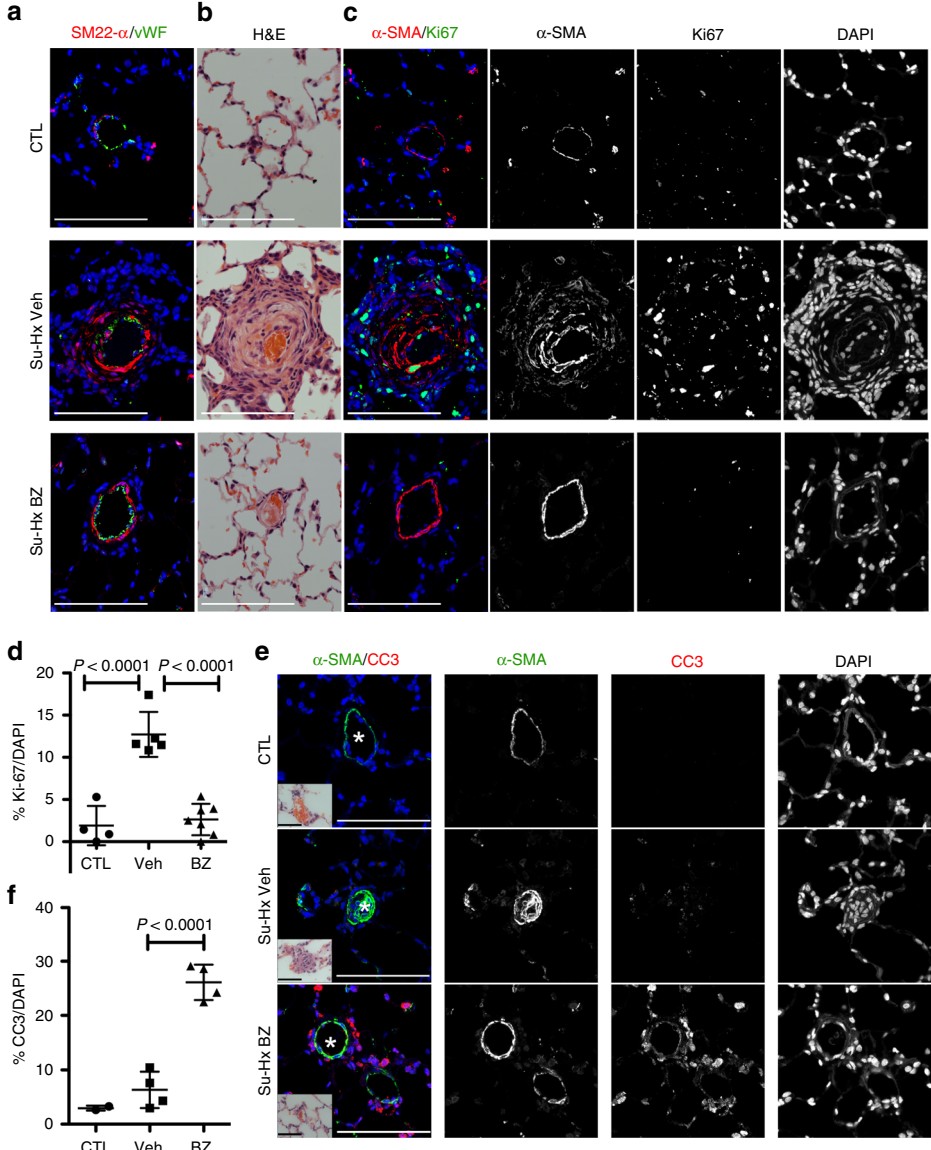

**Fig. 5** Benzarone reduces proliferation and increases apoptosis in the pulmonary vasculature. **a–c** BZ treatment reduces muscularization and cell proliferation in rat lungs subject to the Su-Hx protocol. Serial lung sections were stained with (**a**) antibodies to SM22-α (SMC marker) and von Willebrand factor (vWF; endothelial cell marker), and with the nuclear marker DAPI, (**b**) H&E, (**c**) antibodies to α-SMA and Ki-67 to identify proliferating SMC, and with the nuclear marker DAPI. Scale bar = 100 μm. **d** Quantitation of the percentage of Ki-67-positive cells. Each data point represents at least eight sections per lung. CTL rats maintained in room air n = 4, VEH vehicle-treated animals subject to the Su-Hx protocol n = 5, BZ-treatment protocol n = 7. Data plotted as the mean ± SD, statistical significance determined by one-way ANOVA. **e** BZ induces apoptosis in lungs subject to the Su-Hx protocol. Lung sections were stained with antibodies to cleaved caspase-3 (CC3; an apoptosis marker) and to α-SMA; scale bar = 100 μm. H&E staining of serial sections showing blood vessels indicated by an asterisk are inset; scale = 50 μm. **f** Quantitation of the percentage of apoptotic cells. Each data point represents at least four sections per lung. CTL rats maintained in room air n = 2, VEH vehicle-treated animals subject to the Su-Hx protocol n = 4, BZ-treatment n = 4. Data plotted as the mean ± SD, statistical significance determined by one-way ANOVA. Source Data are provided as a Source Data file

and repair in surviving PAECs through non-EYA3-dependent mechanisms. These trends are similar to those obtained in previous studies with other disease models where BZ treatment reduces the levels of DDR foci[24,29].

Lung tissue from mice was similarly evaluated (Fig. 8). A significant increase in both γ-H2AX-positive and 53BP1-positive cells was measured in pulmonary vascular cells from $Eya3^{+/+}$ mice subject to the Su-Hx protocol (compared with those maintained in room air). In contrast $Eya3^{D262N}$ lung tissue had reduced levels of both markers (Fig. 8), providing genetic evidence that the EYA3 PTP activity plays a role in the assembly and resolution of DDR complexes.

## Discussion

Muscularization of vascular walls[36] and phenotypic alteration of endothelial cells[37,38] contribute to vascular remodeling, the predominant pathological characteristic of PAH. The occlusion of small arterioles caused by the proliferation of phenotypically altered and transdifferentiated PAECs is considered irreversible[34]. But there has been some suggestion that PASMC can shift between proliferative and non-proliferative states raising the possibility that this aspect of vascular remodeling may be reversible[39,40]. The data presented here show that either pharmacological inhibition or genetic loss of the EYA3 protein tyrosine phosphatase can attenuate the development of vascular remodeling and substantially reverse

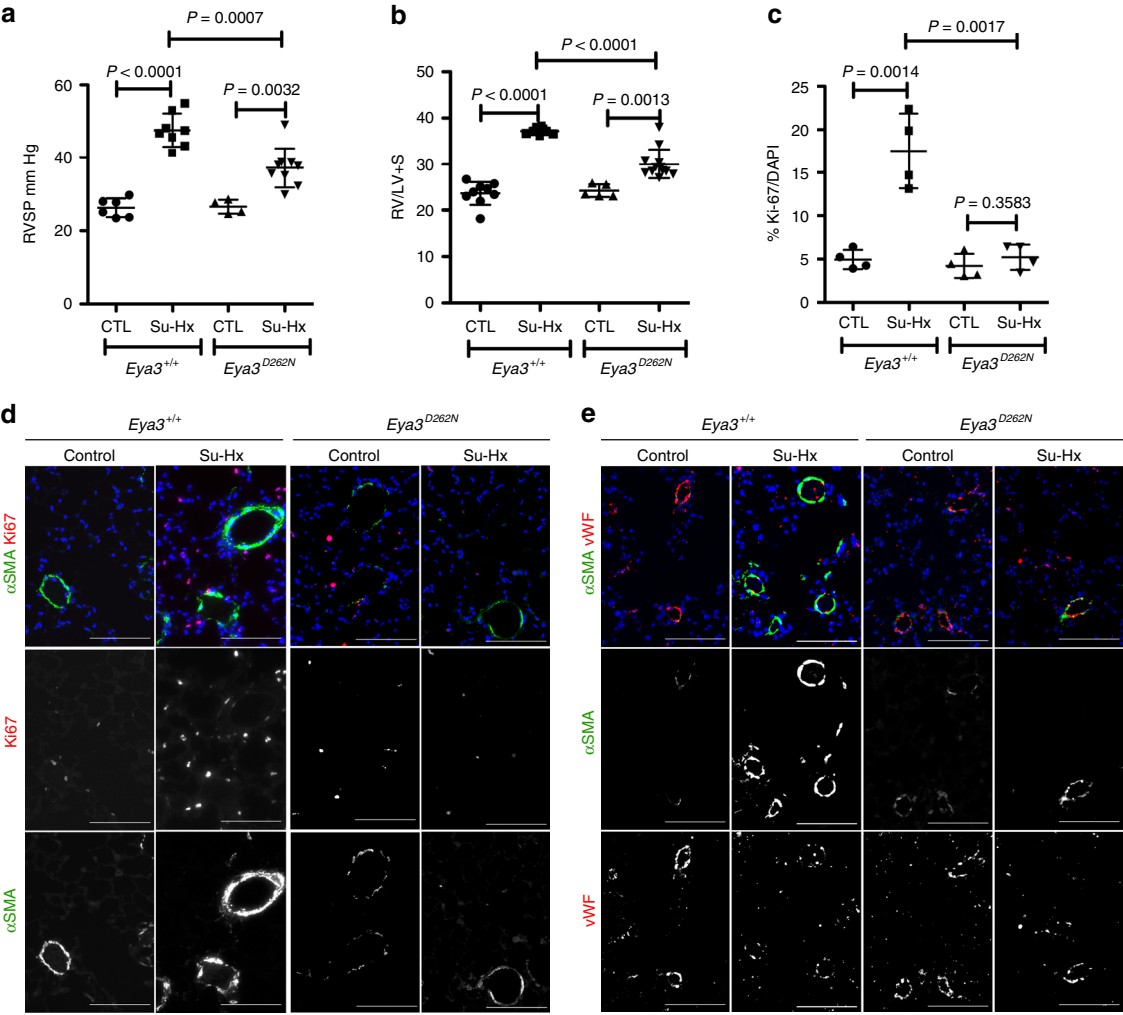

**Fig. 6** Genetic loss of EYA3-PTP activity attenuates hypoxia-induced PH. **a** Right ventricular systolic pressure (RVSP) measured after the Sugen-hypoxia (Su-Hx) protocol is reduced in $Eya3^{D262N}$ mice relative to $Eya3^{+/+}$ control animals. RVSP was measured upon return to room air. Mice maintained in room air served as controls (CTL). Two independent Su-Hx experiments were conducted with a total of 21 C57BL/6J $Eya3^{+/+}$ and 18 $Eya3^{D262N}$ mice. RVSP measurements reported here represent successful catheterization and are plotted as the mean ± SD. Statistical significance was assessed using pairwise $t$-tests. **b** Right ventricular hypertrophy measured after the Su-Hx protocol is reduced in $Eya3^{D262N}$ mice relative to $Eya3^{+/+}$ control animals. Right ventricular hypertrophy (Fulton's index = right ventricular (RV) weight/left ventricle (LV) + interventricular septum weight) measured upon return to room air after the Su-Hx protocol, and in mice maintained in room air for an equivalent period. Data plotted as the mean ± SD. Statistical significance was assessed using pairwise $t$-tests. **c** Quantitation of the percentage of Ki-67-positive cells. Each data point represents at least six sections per lung. For all groups $n = 4$. Data plotted as the mean ± SD, statistical significance determined by pairwise $t$-tests. **d** Cell proliferation and muscularization is reduced in $Eya3^{D262N}$ mice subject to the Su-Hx protocol. Lung sections were stained with antibodies to α-SMA and Ki-67, and with the nuclear marker DAPI. Scale bar = 100 μm. **e** Genetic loss of EYA3-PTP activity reduces muscularization of DPAs but does not affect the endothelial cell layer. Representative sections from lungs of $Eya3^{+/+}$ and $Eya3^{D262N}$ mice either maintained in room air or subject to the Su-Hx protocol were stained for the smooth muscle cell marker α-SMA (green), the endothelial cell marker von Willebrand factor (vWF; red), and the nuclear marker DAPI (blue). Scale bar = 100 μm. Source Data are provided as a Source Data file

established PH in the rat Su-Hx model. The predominant consequence of loss of EYA3 PTP activity reported here is reduced muscularization of small pulmonary arteries.

Pulmonary vascular remodeling is typically the result of temporally aberrant apoptosis, proliferation, and apoptosis-resistance of multiple pulmonary vascular cell types. We analyzed the status of proliferation and apoptosis markers in BZ-treated rat lungs and $Eya3^{D262N}$ mouse lungs after the Su-Hx protocol. As in previous reports[41] the lesions observed in our rat Su-Hx studies are composed of different types of cells. The hypercellular lesions shown in Fig. 5 displayed a gradient of fluorescence intensity for the smooth muscle cell markers α-SMA and SM22-α. Similar staining patterns in other studies led to the suggestion that the

lesions include both SMC and myofibroblasts[42]. In the mouse model, where we observe an increase in RVSP of $Eya3^{+/+}$ mice at the end of the 3-week hypoxia period as in previous reports[32,35], thickening of the walls of pulmonary arterioles was also detected by α-SMA staining (Fig. 6). In both models, at the time points examined here, proliferation was present almost exclusively in non-α-SMA-positive cells. This is consistent with the proposal that increased SMC proliferation is a transient, early phase in vascular remodeling[43–45]. Notably, there was an overall decrease in Ki-67 staining upon either pharmacological or genetic loss of EYA3-PTP activity. Since the experimental strategies used here are not cell-type specific, the cell-autonomous contributions of EYA3 to vascular remodeling remain to be established.

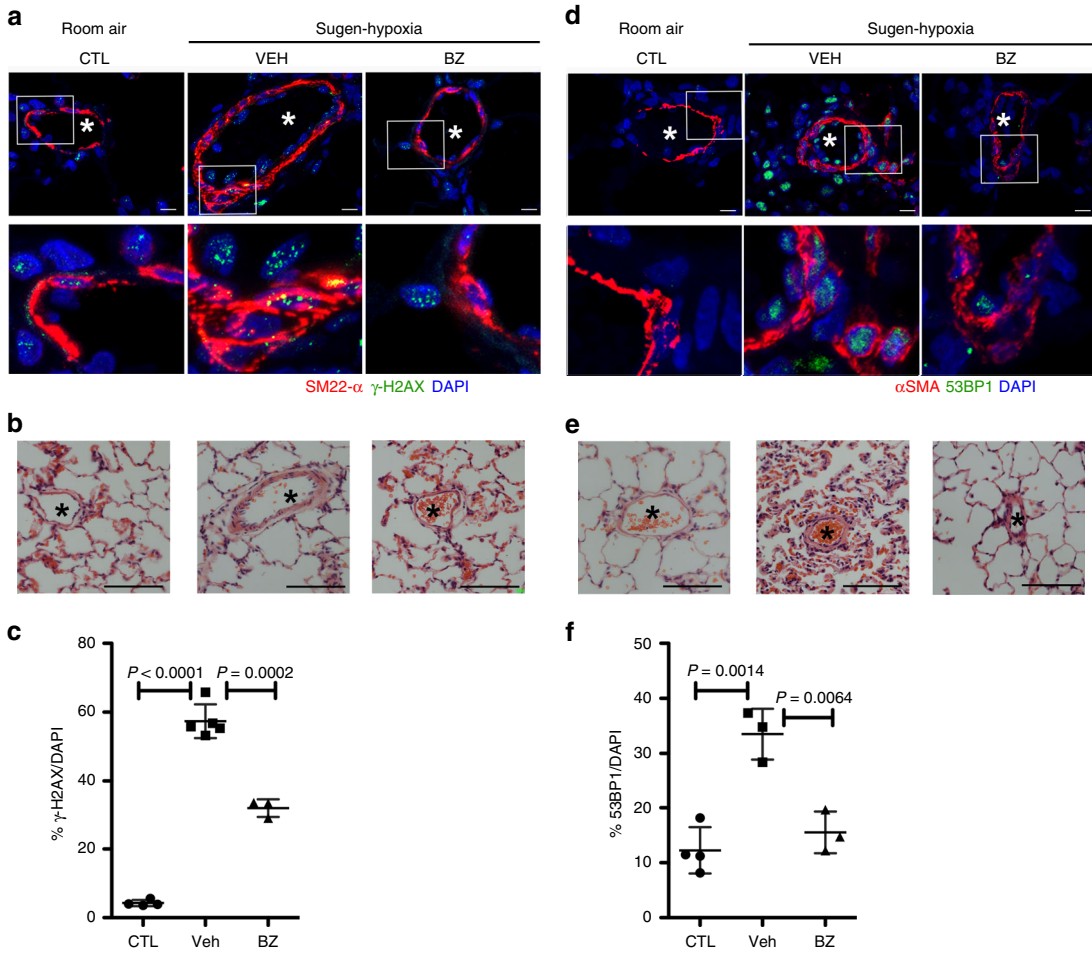

**Fig. 7** EYA-PTP inhibition reduces markers of DNA damage and repair. **a** BZ treatment reduces the formation of DDR foci in rat lungs subject to the Su-Hx protocol. γ-H2AX (green) in rat lung tissue maintained in room air (CTL) and subject to the Su-Hx protocol. γ-H2AX is seen in both SMC (red; SM22-α positive) and non-SMC. DAPI (blue) staining indicates nuclei. Higher magnification of the boxed areas shown in the lower panel. **b** H&E images of serial sections indicating the blood vessels imaged in (**a**). Scale bar = 100 μm. **c** Quantitation of γ-H2AX-positive cells. Tissue sections were analyzed for DNA-damage repair as marked by the presence of γ-H2AX foci. At least four sections per animal were averaged for each data point. CTL rats maintained in room air $n = 4$, VEH vehicle-treated animals subject to the Su-Hx protocol $n = 5$, BZ treatment protocol $n = 3$. Data plotted as the mean ± SD, statistical significance derived from pairwise $t$-tests. **d** BZ treatment reduces the formation of 53BP1 foci in rat lungs subject to the Su-Hx protocol. 53BP1 (green) in rat lung tissue maintained in room air (CTL) and subject to the Su-Hx protocol. Sections were also stained with antibodies toward α-SMA (red) and DAPI (blue). Higher magnification of the boxed areas shown in the lower panel. **e** H&E images of serial sections indicating the blood vessels imaged in (**d**). Scale bar = 100 μm. **f** Quantitation of γ53BP1-positive cells. Tissue sections were analyzed for the assembly of DNA-damage repair as marked by the presence of 53BP1 foci. At least four sections per animal were averaged for each data point. CTL rats maintained in room air $n = 4$, VEH vehicle-treated animals subject to the Su-Hx protocol $n = 3$, BZ treatment protocol $n = 3$. Data plotted as the mean ± SD, statistical significance derived from pairwise $t$-tests. Source Data are provided as a Source Data file

EYA3 has multiple biochemical activities (tyrosine phosphatase, transcriptional activation, and possibly threonine phosphatase[18,19,46–49]). Here we specifically implicate the protein tyrosine phosphatase activity in the development and maintenance of pulmonary vascular remodeling using both genetic and pharmacological methods. We show that the EYA3-PTP activity promotes survival of pulmonary vascular cells in the presence of DNA damage. There is much evidence for higher levels of DNA damage[2,7,50,51], genomic instability[5–7,50], and altered DNA repair mechanisms[3,7] in PAH lungs. The co-existence of these seemingly opposing phenomena is a property that PAH pulmonary vascular cells share with tumor[52] and ageing cells[53]. Error-prone or mutagenic DNA repair processes contribute to genomic instability and to chromosomal abnormalities that promote cell proliferation, while H2AX-dependent DNA repair prevents cell death in the presence of DNA damage. The consequences of DNA damage in pulmonary vascular cells appear to be cell-type

and context-specific. In PAH-PAEC DNA damage is accompanied by increased genomic instability[5,6,50], apoptosis in the early stages of disease, mesenchymal transformation, and release of pro-inflammatory stimuli. PAH-PASMC on the other hand resist apoptosis and can proliferate despite the hostile microenvironment[1,3,7]. Several DDR pathways are elevated in PASMC from PAH patients including the DNA-damage repair factor poly(ADP-ribose) polymerase-1 (PARP-1)[3,7], and the transcription factor FOXM1 that can promote the expression of the DNA repair protein NBS1 (Nijmegen breakage syndrome 1)[30]. Indeed, PARP-1 inhibition can reverse PAH in vivo and the PARP-1 inhibitor Olaparib is in clinical trials as a PAH therapeutic[1,3,7].

Previous studies have shown that the EYA-PTP activity also promotes migration of endothelial cells and angiogenesis[27,28], and that EYA-PTP inhibition is anti-angiogenic[24,29]. However, the loss of EYA3-PTP activity has no detectable impact on the pulmonary

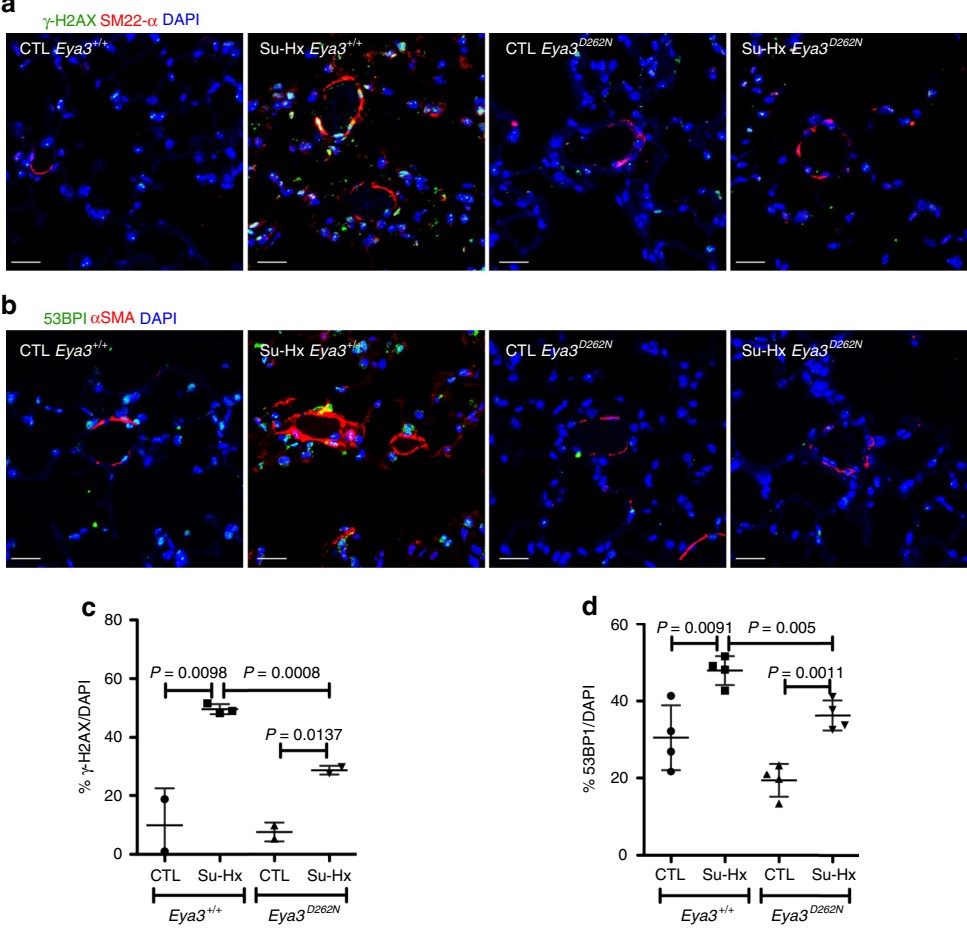

**Fig. 8** Genetic loss of EYA3 PTP activity reduces markers of DNA damage and repair. **a** Genetic loss of EYA3-PTP activity reduces the formation of γ-H2AX foci in lungs of mice subject to the Su-Hx protocol. γ-H2AX (green) and smooth muscle cells (red; SM22-α staining) as seen in lung sections of $Eya3^{+/+}$ and $Eya3^{D262N}$ mice either maintained in room air (CTL) or subject to the Su-Hx protocol. DAPI (blue) staining indicates nuclei; scale bar = 20 μm. **b** Genetic loss of EYA3-PTP activity reduces the formation of 53BP1 foci in lungs of mice subject to the Su-Hx protocol. 53BP1 (green) and smooth muscle cells (red; α-SMA staining) as seen in lung sections of $Eya3^{+/+}$ and $Eya3^{D262N}$ mice either maintained in room air (CTL) or subject to the Su-Hx protocol. DAPI (blue) staining indicates nuclei; scale bar = 20 μm. **c** Quantitation of γ-H2AX-positive cells. Tissue sections were analyzed for DNA-damage repair as marked by the presence of γ-H2AX foci. At least four sections per animal were averaged for each data point. $Eya3^{+/+}$ mice maintained in room air $n = 2$, $Eya3^{+/+}$ mice subject to the Su-Hx protocol $n = 3$, $Eya^{D262N}$ mice maintained in room air $n = 2$, $Eya3^{D262N}$ mice subject to the Su-Hx protocol $n = 2$. Data plotted as the mean ± SD, statistical significance derived from pairwise t-tests. **d** Quantitation of 53BP1-positive cells. At least four sections per animal were averaged for each data point. Data plotted as the mean ± SD ($n = 4$ in each group), statistical significance derived from pairwise t-tests. Source Data are provided as a Source Data file

endothelium in the animal models and at the time points investigated here. This is relevant since the use of anti-angiogenics in the treatment of PAH is complicated by the counter-intuitive observation that VEGF pathway inhibition in animal models contributes to the development of angio-obliterative PAH[31]. It is thought that VEGF-inhibition induces endothelial cell apoptosis during disease initiation[54] and compensatory reprogramming of cellular tyrosine kinase signaling later leads to the emergence of apoptosis-resistant cells (reviewed in ref. [55]).

As with any small molecule inhibitor BZ bears the risk of off-target effects. BZ is a metabolite of a well-known gout drug Benzbromarone and is an inhibitor of xanthine oxidase (XO), an enzyme that generates ROS in the lungs of neonatal rats[56], adult rats[57], and humans[58] with hypoxia-induced PH. Endothelial XO levels are increased by hypoxia both in vitro[59,60] and in vivo[57], and the resulting ROS is an initiating and sustaining factor in PH[57,61–63]. ROS scavengers and the XO inhibitor Allopurinol attenuate the development of PH in rats, and XO inhibition is considered a preferred strategy for targeting ROS in PH[57]. It is

thus possible that BZ-mediated inhibition of XO could also contribute to the attenuated development of PH. However, there is no reported evidence that XO inhibition can reverse pulmonary vascular remodeling in established disease, as seen here with the use of BZ. Further, transgenic mice with a PTP-inactivating amino-acid replacement in EYA3 have reduced arteriolar muscularization and RVSP elevation after hypoxia. The similarity in outcome between chemical inhibition of EYA3-PTP activity and loss of the PTP activity through mutation implicates EYA3 in pulmonary vascular remodeling and attests to the relevance of BZ-mediated EYA-PTP inhibition in the reversal of arteriolar muscularization. More extensive evaluation of functional parameters including measurement of cardiac output, pulmonary vascular resistance and RV function remains necessary.

While BZ provides pharmacological validation of the EYA3-PTP as a target for PH therapeutics, the known hepatic toxicity of this class of compounds[64,65] make it necessary to assess derivatives of BZ that lack hepatotoxicity, or dosage regimens that avert/minimize hepatotoxicity. Notably, we observed no

hepatotoxicity with the dosage regimen used here (Supplementary Fig. 3). Apoptosis in other organs will need to be assessed before clinical translation. Also pertinent to the validation of EYA3-PTP as a target for PH therapeutics, the EYA family of proteins are not widely expressed in adult tissue (reviewed in ref. [19]). EYA3 is the only EYA protein present in human PASMC and PAEC. EYA3-deficient mice have no apparent defects in development, activity, fertility, or life-span[66], supporting the potential safety of therapeutically targeting the EYA3-PTP activity.

In conclusion, our studies support the EYA3-PTP as a contributor to pulmonary vascular remodeling (target-validation) and demonstrate the utility of Benzarone as a tool compound for the development of EYA-PTP inhibitors as effective PAH therapeutics. Clinical translation of these findings would require the completion of critical next steps. A more comprehensive assessment of functional hemodynamics is necessary to evaluate whether the observed reverse remodeling is due to a primary effect on the pulmonary vasculature or is secondary to improved systemic hemodynamics. The cell-autonomous contributions of smooth muscle and endothelial cell EYA3, and other EYA3-modulated molecular mechanisms (in addition to DNA-damage repair) that contribute to vascular remodeling need to be defined. These are the subject of ongoing studies.

## Methods

**Animal studies.** All animal protocols were approved by the Cincinnati Children's Hospital Medical Center IACUC (protocol number IACUC2016-0062), complied with all relevant ethical regulations, and were in accordance with recent recommendations, including randomization and blinded assessments.

For the generation of Eya3$^{D262N}$ mice three sgRNAs that target the intended mutation site (Asp262 to Asn; amino-acid numbering based on AAH63259, D262 refers to the nucleophilic Asp in the DLDET catalytic motif) were selected, according to the off-target scores from the CRISPR Design Tool website (http://www.genome-engineering.org/). Pairs of complementary DNA oligos with compatible overhangs were annealed and cloned into a pX458 vector that carries a U6 promoter to drive sgRNA expression and a ubiquitously expressed promoter to drive Cas9-2A-GFP expression (Addgene plasmid #43138). sgRNA editing activity was evaluated in mouse mK4 cells by the T7E1 assay (New England Biolabs), and compared side-by-side with Tet2 sgRNA that has been shown to edit the mouse genome efficiently. Validated sgRNA and Cas9 mRNA were in vitro transcribed using MEGAshorscript T7 kit and mMESSAGE mMACHINE T7 ULTRA kit (ThermoFisher), respectively, according to manufacturer's instruction. The single-stranded donor oligo that carries homologous arms (>60 nt each) and intended mutations with additional silent mutations to create a new restriction enzyme site was subsequently generated. sgRNA, Cas9 mRNA, and donor ssDNA were mixed at concentration of 50, 100, and 100 ng/μl, respectively, and injected in the cytoplasm of one-cell-stage embryos of C57BL/6J genetic background. Injected embryos were immediately transferred into the ovi-ductal ampulla of pseudo-pregnant CD-1 females. Live born pups were genotyped by PCR and further confirmed by Sanger sequencing. Genotype-confirmed offspring were bred and housed in a vivarium with a 12-h light/dark cycle. All animal studies were approved by the Institutional Animal Care and Use Committees of the Cincinnati Children's Hospital Medical Center, USA. The resulting Eya3$^{D262N}$ mice were then back-crossed to C57BL/6J for >7 generations. C57BL/6J mice were obtained from Jackson Laboratories.

In mouse Su-Hx experiments, 8-week-old C57BL/6J and Eya3$^{D262N}$ (in a C57BL/6J background) were administered weekly sub-cutaneous injections of SU-5416 (20 mg/kg in 0.5% carboxyl methylcellulose sodium, 0.4% polysorbate 80, 0.9% benzyl alcohol while maintained at 10% $O_2$ for 3 weeks. A control group of each genotype was maintained in room air with weekly injections of vehicle. The experiment was conducted two times using age-matched C57BL/6J and Eya3$^{D262N}$ mice. The litters included both male and female animals. In both experiments a percentage of control C57BL/6J mice (3/7 in experiment 1 and 1/5 in experiment 2) died during the hypoxia procedure. There was no mortality among the Eya3$^{D262N}$ mice in both experiments.

In rat Su-Hx experiments, adult (6–8 weeks) male Sprague-Dawley rats (Charles River Laboratories) were injected subcutaneously with 20 mg/kg SU-5416 (in 0.5% carboxyl methylcellulose sodium, 0.4% polysorbate 80, 0.9% benzyl alcohol (Sigma)) and then exposed to 3 weeks of normobaric hypoxia (10% $O_2$). Upon return to room air rats were randomly allocated into four groups: (1) intra-peritoneal injection of 25 mg/ml BZ administered on alternate days for 5 weeks, (2) intra-peritoneal injection of vehicle (peanut oil) administered on alternate days for 5 weeks, (3) intra-peritoneal injection of 25 mg/ml BZ initiated after 5 weeks in room air and administered on alternate days for 3 weeks, (4) intra-peritoneal

injection of vehicle initiated after 5 weeks in room air and administered on alternate days for 3 weeks. A control group was maintained in room air for the entire experiment.

In vivo assessment of right ventricular systolic pressure (RVSP), RV hypertrophy and pulmonary vascular remodeling: After 8 weeks in room air (11 weeks for control animals) all rats were anaesthetized with isoflurane and right ventricular pressures were recorded using right heart catheterizations (FTH-1611B-0018 catheter, Transonic systems, Ithaca, NY). Mice were anaesthetized with isoflurane immediately upon return to room air and right ventricular pressures were recorded using right heart catheterizations (FTH-1211B-0018 catheter).

Hearts were excised and dissected to determine the ratio of right ventricular to left ventricular and septal weight [RV/(LV + S)]. One lobe of the left lung was snap frozen in liquid nitrogen and stored at −80 °C until further analysis. The other lobes were inflated with 4% paraformaldehyde in PBS, dehydrated, embedded, and sectioned. Sections were stained with hematoxylin and eosin (H&E), Masson's Trichrome or Elastica van Gieson (EVG) for analysis and to quantify PA wall thickness. Pulmonary arterial wall thickness was calculated as (total vascular area − lumen area)/luminal diameter.

**Histology.** Immunofluorescence assessment of rat lung sections was performed using goat anti-SM22-α (Abcam, AB10135; 1:500 in Blocking Buffer) followed by Alexa-594 conjugated anti-goat secondary antibody (Molecular Probes), rabbit anti-53BP1 (Cell Signaling #4973S; 1:500 dilution), mouse anti-γH2AX (Millipore 05-636; 1:500 dilution), rabbit anti-Von Willebrand Factor (Abcam AB6994, 1:400 dilution), rabbit anti-Ki-67 (Thermo-scientific MA5-14520, 1:500 dilution) followed by Alexa-488 conjugated anti-rabbit secondary antibody (Molecular Probes).

For immuno-histochemistry anti-EYA3 (Abcam ab95876, 1:100 dilution) anti-α-smooth muscle actin (Thermo Fisher MA5-11547, 1:1000 dilution) antibodies were used. Paraffin slides were deparaffined in xylene, re-hydrated through graded ethanol, and subjected to antigen retrieval with 10 mM citric acid (pH 6). Endogenous peroxidase was blocked with 1.5% $H_2O_2$ in PBS for 10 min. Slides were then incubated with primary antibodies, and biotinylated secondary antibodies. For color development 3,3′-diaminobenzidine (Vector Laboratories, SK-4100) was employed and hematoxylin was used as counterstain.

**Cell culture.** PAEC and PASMC from patients diagnosed with idiopathic PAH were provided by the University of Pennsylvania Cell Center under the Pulmonary Hypertension Breakthrough Initiative (PHBI). Funding for the PHBI is provided by the Cardiovascular Medical Research and Education Fund (CMREF). Control (non-disease) human PAEC (Cat. No. CC-2530) and PASMC (Cat. No. CC-2581) were obtained from Lonza (Allendale, NJ). The use of de-identified human tissue/cells was reviewed by the Cincinnati Children's Hospital Medical Center Institutional Review Board (Study ID:2016-1348) who determined that the studies did not meet the regulatory criteria for research involving human subjects.

For survival experiments (Fig. 2), cells were treated with 200 μM $H_2O_2$ for 1 h. $H_2O_2$ was then withdrawn and the cells allowed to recover in normal culture medium. The percentage of viable cells (relative to untreated controls) were monitored using the WST-8 cell viability assay following the manufacture's protocol (CCK-8 Kit; Dojindo Molecular Technologies, Japan).

The mRNA levels of Eya1, Eya2, Eya3, Eya4 in normal and iPAH patient PASMC and PAEC were analyzed by RT-PCR. Total RNA was purified via PureLink$^{TM}$ RNA Mini Kit (Life Technologies, USA). cDNA was synthesized with PrimeScript$^{TM}$ reagent Kit (Takara, Japan). The sequences of primers used are EYA1 Sense ATGGAAATGCAGGATCT/Anti-sense GGTAGCTGTATGGTG; EYA2 Sense GGACAATGAGATTGAGCGTGT/Anti-sense ATGTCCCCG TGAGAGTAAGGAGT; EYA3 Sense ATGGAAGAAGAGCAAGA/Anti-sense GTTTGGGTTGCCTGAGG; EYA4 Sense CCAGGTCTATGGAAATG/Anti-sense GTTTGAGCTGCTGGTC; GAPDH Sense TTCATTGACCTCAACTAC/Anti-sense CATGGACTGTGGTCATGAG.

EYA3 protein levels in normal and patient PASMCs and PAECs were analyzed by western blot. Antibody dilutions used were 1:2000 anti-EYA3 (Proteintech 21196-1-AP), 1:10,000 pan anti-actin C4 (Seven Hills Bioreagents[67]).

**COMET assays.** Cells were subject to the alkaline COMET assay[68]. Briefly, cells embedded in agarose on glass slides were maintained in lysis buffer (2.5 M NaCl, 100 mM EDTA, 10 mM Tris, 1% sarkosyl, 1% Triton X-100, pH 10; 1 h, 4 °C), neutralized with Tris-buffered EDTA, incubated with alkaline electrophoresis buffer (300 mM NaOH, 1 mM EDTA, pH 12.3) for 30 min to allow DNA to unwind, electrophoresed (40 min, 25 V), slide immersed in 70% ethanol for 5 min, dried at 40 °C for 15 min, and stained with propidium iodide. For quantification comets were imaged with a fluorescent microscope and tail moments calculated using OPENCOMET[69] in ImageJ[70].

**Statistics.** Results presented are the mean + SD (standard deviation). Statistical analyses were performed using Graphpad PRISM version 5.0 for Mac OSX (www.graphpad.com) as described in the legends. Typically a t-test was used when two samples/conditions were compared and ANOVA for more than two groups.

**Reporting summary**. Further information on research design is available in the Nature Research Reporting Summary linked to this article.

## Data availability

The authors declare that all data supporting the findings of this study are available within the paper and its supplementary information files and that all data are available from the corresponding author upon reasonable request. The source data underlying Figs. 1–8 and Supplementary Figs. 2 and 3 are provided as a Source Data file.

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

## Acknowledgements

This work was supported by grants from the NIH National Center for Accelerated Innovation grant NIH-NCAI-14-2-APP-CCHMC-Hegde and the National Cancer Institute grant RO1CA207068 to R.S.H.

## Author contributions

Y.W. and R.N.P. performed all experiments and participated in analyses. J.M. participated in pilot experiments using the rat Sugen-hypoxia protocol. A.Y. and J.D.M. assisted in measurement of right ventricular systolic pressure, W.C.N. provided access to hypoxia chambers for the studies, Y.C.-H., designed guide RNAs and generated mutant mice in the CCHMC transgenic facility, K.A.W.-B. conducted blinded assessments of hepatic toxicity, R.S.H. conceived and designed the project, analyzed data with Y.W. and R.N.P., and wrote the paper. All of the authors have read, edited, and approved the paper.

## Additional information

**Competing interests:** The authors declare no competing interests.

