## [Peer Review File · Nature Communications]

Reviewers' comments:

Reviewer #1 (Remarks to the Author):

The current study interrogates the protein Eyes Absent3 (EYA3) as a target to ameliorate hypertrophic vascular remodeling, a key feature of pulmonary arterial hypertension (PAH) that is causal for right ventricular failure and premature death. The study capitalizes on the role of EYA3 as the branchpoint between survival and apoptosis and tests the hypothesis that inhibition of EYA3 will decrease cell survival, increase apoptosis, and lead to a decrease in vascular remodeling. Experiments show that EYA3 expression is increased in hypertrophied pulmonary vessels from humans with PAH; that increased EYA3 expression is restricted to smooth muscle cells and confers a pro-survival phenotype on these cells; that downregulation or pharmacologic inhibition decreases cell survival; and, in rat and mouse models of pulmonary hypertension with either genetic or pharmacologic inhibition of EYA3 there is a decrease in pulmonary vascular remodeling and pulmonary hypertension. There are some aspects of the study that require attention as figures show that almost all DNA damage occurs outside of vessels/pulmonary artery smooth muscle cells, which raises questions pertaining to the rationale/interpretation of the treatment with BZ affecting smooth muscle cells only. This also raises the point that paracrine effects are important although this is never addressed. The effect of BZ on endothelial cells is also questioned.

1. The Introduction requires major revision to put your study in context of what is known in the disease and correct some inaccuracies about PH as well as remove redundant statements. Comments about vascular remodeling in PH across WHO Groups 1-5 ignores what is known about this phenomenon in different types of PH. In PAH, there is unchecked PASMC proliferation and apoptosis resistance, which is what you study here. While there are some recent pathological studies that show distal pulmonary arterial remodeling does occur in Group 2 PH, this occurs for mechanistically different reasons and there is no evidence that DNA damage occurs here. In fact, there is no evidence that there is DNA damage in Groups 2-5 PH specifically. Therefore, the Introduction needs to be revised to discuss PAH, vascular remodeling in PAH, and what is known about DNA damage/response in Group1 PH.

2. Introduction - Also missing from the Introduction is an overview of what is known about EYA3 in cancer and some introduction to the PH-cancer hypothesis to support rationale for your study.

3. Introduction - Please remove (or revise) the abbreviation PVR. In PH, the term PVR refers to pulmonary vascular resistance, a key hemodynamic measure in the disease.

4. Introduction - What metal is involved in the metal-dependent phosphorylation reaction?

5. A central tenet of your premise is that there is increased oxidant stress in the systems that evokes DNA damage and response and on mechanism of action of benzarone is xanthine oxidase (ROS-generating enzyme). At the least, oxidant stress should be measured in cells/pulmonary vasculature in cell experiments and in vivo studies.

6. Studies characterizing PH in vivo are rudimentary and missing some important hemodynamics and RV characterizations aren't provided. The data for arterial blood pressure, heart weight, body weight, cardiac output, pulmonary vascular resistance should be given. It's also typical to provide information on RV function.

7. Benzarone administration is associated with hepatic toxicity. Liver function tests should be done/shown as evidence that the drug is well tolerated. Since benzarone inhibits EYA1 and EYA3, it is necessary to show that this is tolerated in the PH models by random sampling of organs that express either and looking for evidence of apoptosis. This speaks to feasibility of translating this therapy to humans with PAH.

8. Even though EYA3 expression is not increased in PAECs from PAH patients, it is expressed. This suggests that inhibition with BZ will cause apoptosis in PAECs as these cells are known to express markers of DNA damage (g-H2AX). Studies should be done with these cells as apoptosis of ECs would not benefit vascular remodeling.

9. Fig. 1 – comments that EYA3 expression is increased in smooth muscle cell aren't quite supported by the IHC studies that are shown. While it appears that this is correct, you need to do double IF studies to show that there is co-expression. It also appears that there is EYA3 expression outside the smooth muscle cells.

-

10. Fig. 2 – Following transfection with EYA3 shRNA it's expected that cells with decreased/near normal EYA3 levels would don't behave the same as non-transfected PSMCs (decreased cell survival), yet they don't. Why? An additional and needed control here is to transfect cells with a construct that mutates the threonine phosphatase and one that mutates the transcriptional binding site. This would confirm that it's only the Y phosphatase activity that is important.

11. Fig. 3 – description of results in the text is misleading (or incorrect) as written. Can't have an increase in RVSP that's 28% lower and 70% lower when they are nearly identical. It would be better

to just provide the numbers for the reader to review and not report as done, which magnifies the treatment effect.

12. Fig. 4 – you measure “occluded vessels” but none of the vessels you show are occluded. They contain circulating blood cells in the lumen. Typically, it is reported as muscularized vessels of different calibers. This should be revised and a proper analysis done. Or show actual occluded vessels, which is unlikely given the pressures you report.

13. Fig. 5a – the majority of your proliferating cells are outside the vessel. The representative sections in the CC3 sections for vehicle-treated animals don’t show convincing remodeled distal pulmonary arterioles. Moreover, the CC3 positive cells don’t line up with the α -SMA cells suggesting that BZ is having an effect on adventitial cells, not the smooth muscle cells. How do you reconcile this?

14. Fig. 6c – the images appear to be mislabeled (CTL vs Su-HX for wild-type) – there are no hypertrophied vessels (increased α -SMA) or Ki67 positive cells. These images also need an H&E panel to show the reader what structures are in the image.

15. Fig 6f – While the purported aim of this figure is to show that EYA3 absence has no effect on PAECs in PH, the images raise concern about your models. Your WT-Su-Hx vessels label only for ECs but no α -SMA. By contrast, the CTLs here show only α -SMA in vessels and no labeling for ECs. This doesn’t make sense. Because the controls don’t make sense, it isn’t clear how to interpret findings from the knockout animals.

16. Fig. 7 – the majority of the DNA damaged cells (stain positive for γ -H2AX) are outside of the medial layer of the vessel. How is this explained? What cells are involved? Are these vascular or parenchymal?

17. Fig. 8 – same comment as #16.

188. Discussion should be streamlined to discuss within the context of PH.

Reviewer #2 (Remarks to the Author):

In the manuscript by Wang et al., the authors examine the role of the Eya3 Tyrosine phosphatase in pulmonary hypertension. The authors demonstrate that Eya3 levels are increased in pulmonary arterial smooth muscle cells isolated from patients with Pulmonary arterial hypertension (PAH). Using both a novel inhibitor of the Eya3 tyrosine phosphatase activity (benzarone) as well as a mouse model in which the tyrosine phosphatase activity of Eya3 is knocked out via a point mutation, the authors demonstrate a role for the Eya3 Tyr phosphatase activity in the development of pulmonary hypertension and in the vascular remodeling that occurs in pulmonary hypertension. These experiments suggest that targeting of the Eya3 phosphatase activity may be a potential druggable avenue for pulmonary hypertension, which is an exciting finding. In general, the data within this paper are clearly presented and are correctly interpreted to demonstrate a key role for Eya3 in this process. The manuscript could be improved if the following comments were addressed:

1. The authors state that Eya3 levels are not elevated in PAH-PAEC relative to normal PAEC cells but the PAH-PAEC is very underloaded, and so it is hard to come to that conclusion from the data shown in Fig. 1.
2. Two KD constructs (shRNAs targeting different regions of Eya3) should be used for the experiments in Fig. 2 rather than one KD construct to ensure that the effects observed are not off target.
3. It would be interesting to see if Benzarone would still confer an effect in the shEya3 cells (in experiments in fig. 2). If it did- one may be concerned that the effects observed on survival in this context may be more through targeting of XO.
4. In figure 2 the authors use H₂O₂ to mimic hypoxia induced death in PASMC from normal or PAH patients. It would be nice to see this assay under true hypoxic conditions and to also see the DNA damage in this setting (or DNA repair by examining gH2AX foci).
5. More discussion of the role of Eya3 in the initiation vs maintenance of PAH may be warranted as the treatment strategy appears to be more effective than the prevention strategy.
6. While the authors argue that the effects of Eya3 Tyr phosphatase activity are specific to SMCs, given the expression of Eya3 in SMCs and ECs, and the fact that the genetic model and the benzarone would affect the activity in both cell types, is it not possible that phenotypes are observed due to effects in multiple cellular compartments?

Reviewer #3 (Remarks to the Author):

In this study, Yuhua et al show that a small molecule inhibitor can be used to reverse pulmonary hypertension in an experimental model. Similarly, EYA transgenic mice with a mutation at its phosphatase domain are also protected from pulmonary hypertension. The mice phenotype is interesting. My major criticism is that it is unclear whether the small compound Benzarone functions through EYA or not, and what is the role of DNA damage response in their models. Could it be through other functions of EYA instead of the DDR? The study does not elucidate the mechanism of the observed effects. My point by point criticism follows:

1. Benzarone is a nonpurine xanthine oxidase inhibitor but never approved for use in the United States because of concerns over reports of acute liver injury and deaths with its use (PMID: 15799034). I do not see the utility of this study given the very significant side effect of the drug the author's used in this study.
2. The manuscript lacks proof that the drug BZ targets the phosphatase EYA3. Experiments that definitively show that BZ acts through inhibiting EYA in vivo are required to strengthen the manuscript.
3. The studies are all correlational in my opinion. Mechanistic studies are lacking.
4. Inhibiting EYA is not known to decrease gammaH2AX signals. This raised the question whether BZ or EYAD262N really affects pulmonary hypertension through regulating the DNA damage response pathway.
5. Is EYA upregulated in Su-Hx models?
6. The age of the patients that were the sources of the normal and abnormal tissue is not clear. Since age can affect lung tissue, the control and disease tissue need to be of approximately the same age.
7. Figure 2 and Su-Hx models treated with BZ, the authors should examine DNA damage, gammaH2AX and pY-H2AX levels.
8. The images in figure 5 appear to be of different magnification (a vs c) even though the scale bar is shown to be the same.
9. Figure 8, some gammaH2AX signals seems like debris, not cells.

Minor comments:

1. Please provide H&E images corresponding to each immunofluorescence imaging for better interpretation of the data.
2. The manuscript needs to be proofread for typos.
3. Please provide scale bars for all immunofluorescence images. It will also be nice to see further magnified images in addition to the ones already provided.

We thank the reviewers for very thoughtful and detailed comments on the manuscript. In this revision every one of these comments has been addressed with additional experiments, modifications in data presentation and/or additional discussion as appropriate. We hope that this version of the manuscript has addressed the reviewer's concerns. Responses to individual comments are shown in *blue italics* below.

Reviewer #1 (Remarks to the Author):

The current study interrogates the protein Eyes Absent3 (EYA3) as a target to ameliorate hypertrophic vascular remodeling, a key feature of pulmonary arterial hypertension (PAH) that is causal for right ventricular failure and premature death. The study capitalizes on the role of EYA3 as the branchpoint between survival and apoptosis and tests the hypothesis that inhibition of EYA3 will decrease cell survival, increase apoptosis, and lead to a decrease in vascular remodeling. Experiments show that EYA3 expression is increased in hypertrophied pulmonary vessels from humans with PAH; that increased EYA3 expression is restricted to smooth muscle cells and confers a pro-survival phenotype on these cells; that downregulation or pharmacologic inhibition decreases cell survival; and, in rat and mouse models of pulmonary hypertension with either genetic or pharmacologic inhibition of EYA3 there is a decrease in pulmonary vascular remodeling and pulmonary hypertension. There are some aspects of the study that require attention as figures show that almost all DNA damage occurs outside of vessels/pulmonary artery smooth muscle cells, which raises questions pertaining to the rationale/interpretation of the treatment with BZ affecting smooth muscle cells only. This also raises the point that paracrine effects are important although this is never addressed. The effect of BZ on endothelial cells is also questioned.

1. The Introduction requires major revision to put your study in context of what is known in the disease and correct some inaccuracies about PH as well as remove redundant statements. Comments about vascular remodeling in PH across WHO Groups 1-5 ignores what is known about this phenomenon in different types of PH. In PAH, there is unchecked PSMC proliferation and apoptosis resistance, which is what you study here. While there are some recent pathological studies that show distal pulmonary arterial remodeling does occur in Group 2 PH, this occurs for mechanistically different reasons and there is no evidence that DNA damage occurs here. In fact, there is no evidence that there is DNA damage in Groups 2-5 PH specifically. Therefore, the Introduction needs to be revised to discuss PAH, vascular remodeling in PAH, and what is known about DNA damage/response in Group1 PH.

The opening paragraphs of the introduction have been modified to further emphasize that the mechanisms described here refer to Group 1 PH/PAH.

2. Introduction - Also missing from the Introduction is an overview of what is known about EYA3 in cancer and some introduction to the PH-cancer hypothesis to support rationale for your study.

Both of these are now covered in paragraphs 2 and 3 of the Introduction.

3. Introduction - Please remove (or revise) the abbreviation PVR. In PH, the term PVR refers to pulmonary vascular resistance, a key hemodynamic measure in the disease.

We have removed the term PVR.

4. Introduction - What metal is involved in the metal-dependent phosphorylation reaction?

The divalent metal is typically Mg²⁺. The metal-dependence of in vitro EYA3 PTP activity has been covered in previous studies (referenced in the introduction).

5. A central tenet of your premise is that there is increased oxidant stress in the systems that evokes DNA damage and response and on mechanism of action of benzarone is xanthine oxidase (ROS-generating

enzyme). At the least, oxidant stress should be measured in cells/pulmonary vasculature in cell experiments and in vivo studies.

To clarify, our central hypothesis is not that Benzarone is acting through xanthine oxidase in this context. We propose that the Eyes Absent protein (EYA3) protein tyrosine phosphatase contributes to vascular remodeling through its ability to promote survival of DNA damaged cells facing a survival-versus-apoptosis decision (as stated in the Abstract and Introduction). The possibility that XO activity is affected is covered in the discussion.

6. Studies characterizing PH in vivo are rudimentary and missing some important hemodynamics and RV characterizations aren't provided. The data for arterial blood pressure, heart weight, body weight, cardiac output, pulmonary vascular resistance should be given. It's also typical to provide information on RV function.

Body and heart weight information are provided in the legend to Figure 3. The goal of the present exploratory pre-clinical study was to use genetic and pharmacological approaches to investigate whether the EYA3 PTP activity contributes to pulmonary vascular remodeling. As the reviewer rightly points out, more complete hemodynamic assessment is a necessary component of pre-clinical studies and is the subject of ongoing investigations. This limitation is acknowledged in the Conclusions section page 12.

7. Benzarone administration is associated with hepatic toxicity. Liver function tests should be done/shown as evidence that the drug is well tolerated. Since benzarone inhibits EYA1 and EYA3, it is necessary to show that this is tolerated in the PH models by random sampling of organs that express either and looking for evidence of apoptosis. This speaks to feasibility of translating this therapy to humans with PAH.

We measured serum Alanine Transaminase activity as an indicator of liver toxicity in the rat Sugden-hypoxia experiments; ALT levels in rats treated with vehicle was compared to those treated with BZ. The results, showing no change in ALT activity upon BZ treatment, are presented in Supplementary Figure 3.

Furthermore, blinded analyses of liver sections were conducted by a pathologist. Livers from treated and control animals had only minimal liver inflammation in a rare lobular focus or rare portal tracts. No necrosis, fibrosis or neoplasia was present to indicate toxic injury.

Although no evidence of systemic toxicity was observed in the tissues examined here, the potential induction of apoptosis in other organs as a result of BZ treatment and the accumulation of DNA damage remains a concern and would need to be monitored before clinical translation of the present findings. This manuscript serves as target-validation and this is emphasized in the Conclusions section "While BZ provides pharmacological validation of the EYA3-PTP as a target for PH therapeutics, the known hepatic toxicity of this class of compounds¹⁻³ make it necessary to assess derivatives of BZ that lack hepatotoxicity, or dosage regimens that avert/minimize hepatotoxicity. Notably, we observed no hepatotoxicity with the dosage regimen used here (Supplementary Figure 3), although it remains necessary to examine other organs for evidence of apoptosis before clinical translation."

8. Even though EYA3 expression is not increased in PAECs from PAH patients, it is expressed. This suggests that inhibition with BZ will cause apoptosis in PAECs as these cells are known to express markers of DNA damage (g-H2AX). Studies should be done with these cells as apoptosis of ECs would not benefit vascular remodeling.

We have conducted survival experiments on PAEC as suggested by the reviewer and now present them in Supplementary Fig. 2. Interestingly, unlike in the case of PSMC, PAEC survival after oxidative stress is not affected by BZ treatment. Furthermore, PAH-PAEC are similar to normal PAEC in their ability to survive exogenous peroxide.

The reviewer rightly points out that PAECs are known to stain positive for γ -H2AX, a marker of DNA damage. Based on our observations we report that γ -H2AX staining is seen in both PAEC and PSMC in the

Su-Hx model (Fig. 7, 8), and is reduced upon loss of EYA3-PTP activity. At the time-points measured here there is little loss/apoptosis of endothelial cells, but we see substantial reduction in muscularization with the loss of EYA3-PTP activity. The discussion refers to these observations and provides context on page 11: “The consequences of DNA damage in pulmonary vascular cells appear to be cell-type and context-specific. In PAH-PAEC DNA damage is accompanied by increased genomic instability, apoptosis in the early stages of disease, mesenchymal transformation, and release of pro-inflammatory stimuli. PAH-PASMC on the other hand resist apoptosis and can proliferate despite the hostile microenvironment.”

9. Fig. 1 – comments that EYA3 expression is increased in smooth muscle cell aren't quite supported by the IHC studies that are shown. While it appears that this is correct, you need to do double IF studies to show that there is co-expression. It also appears that there is EYA3 expression outside the smooth muscle cells.-

There is EYA3 in non-smooth muscle cells (as stated in Figure 1 and in the Results section). Our data shows that the levels of EYA3 are elevated in PAH-PASMC relative to normal controls. Such an elevation is not consistently seen in PAEC. We show α -SMA and EYA3 staining in serial sections (Fig. 1). We have tried double IF staining, but this was technically challenging in these human tissue samples.

10. Fig. 2 – Following transfection with EYA3 shRNA it's expected that cells with decreased/near normal EYA3 levels would don't behave the same as non-transfected PASMCs (decreased cell survival), yet they don't. Why? An additional and needed control here is to transfect cells with a construct that mutates the threonine phosphatase and one that mutates the transcriptional binding site. This would confirm that it's only the Y phosphatase activity that is important.

L10 and L85 with decreased EYA3 levels (L10-shEYA3, L85-shEYA3) do show decreased cell survival relative to control L10 or L85 cells (Fig. 2) as predicted by the reviewer.

The Thr-phosphatase activity of the EYAs appears to be not intrinsic but rather mediated through interaction with PP2A-B55 α (Zhang et al Nat. Communications (2018) 9:1047). Examination of the role of this interaction and the associated Thr-phosphatase activity/pathway is the subject of separate investigations.

There is no single mutation that can abrogate the transactivation potential of the EYAs. If “transcriptional binding site” refers to the binding site of the EYA-SIX complex on DNA, no such site relevant to this project has been identified.

The use of BZ as a chemical inhibitor of the PTP activity alone provides support for the contribution of the PTP activity to the survival of SMC under oxidative stress. This conclusion is also supported by the in vivo studies using a single point mutation that only abrogates the PTP activity.

11. Fig. 3 – description of results in the text is misleading (or incorrect) as written. Can't have an increase in RVSP that's 28% lower and 70% lower when they are nearly identical. It would be better to just provide the numbers for the reader to review and not report as done, which magnifies the treatment effect.

This section (page 7) has been altered as requested.

12. Fig. 4 – you measure “occluded vessels” but none of the vessels you show are occluded. They contains circulating blood cells in the lumen. Typically, it is reported as muscularized vessels of different calibers. This should be revised and a proper analysis done. Or show actual occluded vessels, which is unlikely given the pressures you report.

This figure and accompanying text have been modified to report muscularized vessels of diameter less than 100 μ m.

13. Fig. 5a – the majority of your proliferating cells are outside the vessel. The representative sections in the CC3 sections for vehicle-treated animals don't show convincing remodeled distal pulmonary arterioles. Moreover, the CC3 positive cells don't line up with the α -SMA cells suggesting that BZ is having an effect on adventitial cells, not the smooth muscle cells. How do you reconcile this?

The reviewer rightly points out that proliferation and apoptosis are not restricted to SMC. This observation is discussed on pages 10-11:

“Pulmonary vascular remodeling is typically the result of temporally aberrant apoptosis, proliferation, and apoptosis-resistance of multiple pulmonary vascular cell types. We analyzed the status of proliferation and apoptosis markers in BZ-treated rat lungs and $Eya3^{D262N}$ mouse lungs after the Su-Hx protocol. As in previous reports⁴⁵ the lesions observed in our rat Su-Hx studies are composed of different types of cells. The hypercellular lesions shown in Fig. 5 showed a gradient of fluorescence intensity for the smooth muscle cell markers α -SMA and SM22- α . Similar staining patterns in other studies led to the suggestion that the lesions include both SMC and myofibroblasts⁴⁶. In the mouse model, where we observe an increase in RVSP of $Eya3^{+/+}$ mice at the end of the 3-week hypoxia period as in previous reports^{38, 41}, thickening of the walls of pulmonary arterioles was also detected by α -SMA staining (Fig. 6). In both models, at the time-points examined here, proliferation was present almost exclusively in non- α -SMA positive cells. This is consistent with the proposal that increased SMC proliferation is a transient, early phase in vascular remodeling^{47, 48, 49}. Notably, there was an overall decrease in Ki-67 staining upon either pharmacological or genetic loss of EYA3-PTP activity. Since the experimental strategies used here are not cell-type specific the cell-autonomous contributions of EYA3 to vascular remodeling remain to be established.”

14. Fig. 6c – the images appear to be mislabeled (CTL vs Su-HX for wild-type) – there are no hypertrophied vessels (increased α -SMA) or Ki67 positive cells. These images also need an H&E panel to show the reader what structures are in the image.

We thank the reviewer very much for picking up this error in labeling. Fig. 6 has been corrected and new images including the requested H&E panel are presented.

15. Fig 6f – While the purported aim of this figure is to show that EYA3 absence has no effect on PAECs in PH, the images raise concern about your models. Your WT-Su-Hx vessels label only for ECs but no α -SMA. By contrast, the CTLs here show only α -SMA in vessels and no labeling for ECs. This doesn't make sense. Because the controls don't make sense, it isn't clear how to interpret findings from the knockout animals. *This is a consequence of the flipped label panel (as in Fig. 6c and pointed out by the reviewer in point 14 above). These labelling panels have been corrected and now rightly reflect our observation that EYA3 absence has no detectable effect on PAECs in PH.*

16. Fig. 7 – the majority of the DNA damaged cells (stain positive for γ -H2AX) are outside of the medial layer of the vessel. How is this explained? What cells are involved? Are these vascular or parenchymal?

17. Fig. 8 – same comment as #16.

In our experiments DNA damage repair complexes were present in both smooth muscle & endothelial cells, and in cell populations surrounding vascular lesions in Su-Hx lung tissue. In both BZ-treated rat lungs and $Eya3^{D262N}$ mouse lung tissue fewer γ -H2AX- and 53BP1-positive cells were seen, supporting a role for EYA3-PTP dependent DNA damage repair in pulmonary vascular remodeling. The specific contributions of individual pulmonary vascular cell types to DDR-associated vascular remodeling remains to be established. Both BZ treatment and the genetic model ($Eya3^{D262N}$) we have used are not cell type specific.

188. Discussion should be streamlined to discuss within the context of PH.

We have streamlined the discussion. Comments related to EYA3 and DDR have been retained as they are pertinent to this project.

Reviewer #2 (Remarks to the Author):

In the manuscript by Wang et al., the authors examine the role of the Eya3 Tyrosine phosphatase in pulmonary hypertension. The authors demonstrate that Eya3 levels are increased in pulmonary arterial smooth muscle cells isolated from patients with Pulmonary arterial hypertension (PAH). Using both a novel inhibitor of the Eya3 tyrosine phosphatase activity (benzarone) as well as a mouse model in which the tyrosine phosphatase activity of Eya3 is knocked out via a point mutation, the authors demonstrate a role for the Eya3 Tyr phosphatase activity in the development of pulmonary hypertension and in the vascular remodeling that occurs in pulmonary hypertension. These experiments suggest that targeting of the Eya3 phosphatase activity may be a potential druggable avenue for pulmonary hypertension, which is an exciting finding. In general, the data within this paper are clearly presented and are correctly interpreted to demonstrate a key role for Eya3 in this process. The manuscript could be improved if the following comments were addressed:

1. The authors state that Eya3 levels are not elevated in PAH-PAEC relative to normal PAEC cells but the PAH-PAEC is very underloaded, and so it is hard to come to that conclusion from the data shown in Fig. 1.

An additional Western blot and quantitation is in Fig. 1.

2. Two KD constructs (shRNAs targeting different regions of Eya3) should be used for the experiments in Fig. 2 rather than one KD construct to ensure that the effects observed are not off target.

Results using additional shEYA3 are now presented in Supplementary Fig. 2.

3. It would be interesting to see if Benzarone would still confer an effect in the shEya3 cells (in experiments in fig. 2). If it did- one may be concerned that the effects observed on survival in this context may be more through targeting of XO.

Benzarone treatment of L10-shEYA3 and L85-shEYA3 cells did not reduce survival relative to vehicle controls (data shown in Fig. 2e, f and Supplementary Figure 2).

4. In figure 2 the authors use H₂O₂ to mimic hypoxia induced death in PASMCM from normal or PAH patients. It would be nice to see this assay under true hypoxic conditions and to also see the DNA damage in this setting (or DNA repair by examining gH2AX foci).

Figure 2 is intended to mimic oxidative-stress (ROS) that is typically a result of the chronic hypoxia and inflammation seen in PAH. Other studies use DNA damaging agents such as etoposide to mimic the DNA damage seen in PASMCM¹⁷. The DNA damage response caused by hypoxia alone can have very different characteristics, including the lack of detectable DNA damage and repression of repair¹⁸. Additionally, in Supplementary Fig. 2 we show that the peroxide treatment used in Figure 2 induces DNA damage as measured by γ -H2AX formation.

5. More discussion of the role of Eya3 in the initiation vs maintenance of PAH may be warranted as the treatment strategy appears to be more effective than the prevention strategy.

While a small difference between treatment and prevention protocols is seen in the RVSP and Fulton's index, these differences are not significant enough to emphasize in this manuscript.

6. While the authors argue that the effects of Eya3 Tyr phosphatase activity are specific to SMCs, given the expression of Eya3 in SMCs and ECs, and the fact that the genetic model and the benzarone would affect the activity in both cell types, is it not possible that phenotypes are observed due to effects in multiple cellular compartments?

The manuscript has been edited at several points to clarify this point. Our data show that EYA3 is expressed in multiple cell types within PAH lesions. However, the levels of EYA3 appear to be most elevated in PAH-PASMC relative to normal PASMC. The effect of both benzarone and the EYA3 mutation are not cell-type specific. The consequences of both of these perturbations include reduced muscularization of small pulmonary arteries as well as reduced proliferation and markers of DNA damage response in multiple pulmonary vascular cell types. The cell-autonomous role(s) of EYA3 in pulmonary vascular remodeling are the subject of ongoing investigation.

Reviewer #3 (Remarks to the Author):

In this study, Yuhua et al show that a small molecule inhibitor can be used to reverse pulmonary hypertension in an experimental model. Similarly, EYA transgenic mice with a mutation at its phosphatase domain are also protected from pulmonary hypertension. The mice phenotype is interesting. My major criticism is that it is unclear whether the small compound Benzarone functions through EYA or not, and what is the role of DNA damage response in their models. Could it be through other functions of EYA instead of the DDR? The study does not elucidate the mechanism of the observed effects. My point by point criticism follows:

1. Benzarone is a nonpurine xanthine oxidase inhibitor but never approved for use in the United States because of concerns over reports of acute liver injury and deaths with its use (PMID: 15799034). I do not see the utility of this study given the very significant side effect of the drug the author's used in this study.

The hepatotoxicity of Benzarone is acknowledged and discussed in the manuscript. The purpose of this study was to provide target-validation of EYA3 as a therapeutic target. In this regard we show the effectiveness of loss of EYA3-PTP activity both genetically and through the use of Benzarone. The similarity in outcome with these two approaches supports the hypothesis that EYA3-PTP activity contributes to vascular remodeling.

As stated in the discussion, further development of BZ derivatives or dosing regimens that minimize hepatotoxicity would be warranted before BZ is considered in the clinic. Notably, in the studies reported here no detectable hepatotoxicity was observed (Supplementary Fig. 3).

2. The manuscript lacks proof that the drug BZ targets the phosphatase EYA3. Experiments that definitively show that BZ acts through inhibiting EYA in vivo are required to strengthen the manuscript. *The ability to BZ to inhibit the EYA3 PTP activity has previously been established using recombinant EYA3¹⁹,²⁰, in cellular assays^{19, 20}, and in vivo^{21, 22}.*

The similarity in outcome using either BZ or a single amino acid mutation of EYA3 that results in a loss of PTP activity strongly argues for an EYA3-PTP-dependent mechanism of action of BZ in the present context. Furthermore, in Fig. 2 (and Supp. Fig. 2) we show that BZ has no effect on PASMC in which EYA3 is knocked-down. Despite this data, the issue of off-target effects remains a concern for BZ (as for any small molecule therapeutic). This concern is discussed on page 12.

3. The studies are all correlational in my opinion. Mechanistic studies are lacking.

Extensive mechanistic studies linking EYA3 to DNA damage repair and cell survival have been reported elsewhere (and referenced in the manuscript). The present report seeks to provide a first link between EYA3 PTP activity and pulmonary vascular remodeling and establishes genetic and pharmacological target validation of EYA3 in this disease state. These studies use experimental intervention to establish cause-and-effect.

4. Inhibiting EYA is not known to decrease gammaH2AX signals. This raised the question whether BZ or EYAD262N really affects pulmonary hypertension through regulating the DNA damage response pathway.

Inhibiting the EYA-PTP activity reduces the γ -H2AX signal in multiple contexts including proliferative retinopathy²², tumor angiogenesis²¹, and in tumor cells²¹.

5. Is EYA upregulated in Su-Hx models?

Fig. 3 now includes a panel showing that EYA3 levels are elevated in the pulmonary vasculature of Sugen-hypoxia rats.

6. The age of the patients that were the sources of the normal and abnormal tissue is not clear. Since age can affect lung tissue, the control and disease tissue need to be of approximately the same age.

This information is included in the legend.

7. Figure 2 and Su-Hx models treated with BZ, the authors should examine DNA damage, gammaH2AX and pY-H2AX levels.

γ -H2AX staining has been monitored in PASMC (Supp. Fig. 2) and in the Su-Hx models (Fig. 7, 8). The available pY-H2AX antibodies do not work reliably on tissue samples.

8. The images in figure 5 appear to be of different magnification (a vs c) even though the scale bar is shown to be the same.

Thank-you for pointing out this error. This, and other, figures have been entirely re-created to include the H&E images and scale bars requested below.

8. Figure 8, some gammaH2AX signals seems like debris, not cells.

γ -H2AX staining can be seen in cells as either foci within the nucleus or as pan-nuclear staining.

Minor comments:

1. Please provide H&E images corresponding to each immunofluorescence imaging for better interpretation of the data.

2. The manuscript needs to be proofread for typos.

3. Please provide scale bars for all immunofluorescence images. It will also be nice to see further magnified images in addition to the ones already provided.

All of these minor comments have been addressed.

References:

1. Jansen TL, Reinders MK, van Roon EN and Brouwers JR. Benzbromarone withdrawn from the European market: another case of "absence of evidence is evidence of absence"? *Clin Exp Rheumatol*. 2004;22:651.
2. Kaufmann P, Torok M, Hanni A, Roberts P, Gasser R and Krahenbuhl S. Mechanisms of benzarone and benzbromarone-induced hepatic toxicity. *Hepatology*. 2005;41:925-35.
3. Lee MH, Graham GG, Williams KM and Day RO. A benefit-risk assessment of benzbromarone in the treatment of gout. Was its withdrawal from the market in the best interest of patients? *Drug safety : an international journal of medical toxicology and drug experience*. 2008;31:643-65.
4. Aldred MA, Comhair SA, Varella-Garcia M, Asosingh K, Xu W, Noon GP, Thistlethwaite PA, Tuder RM, Erzurum SC, Geraci MW and Coldren CD. Somatic chromosome abnormalities in the lungs of patients with pulmonary arterial hypertension. *Am J Respir Crit Care Med*. 2010;182:1153-60.
5. Drake KM, Federici C, Duong HT, Comhair SA, Erzurum SC, Asosingh K and Aldred MA. Genomic stability of pulmonary artery endothelial colony-forming cells in culture. *Pulm Circ*. 2017;7:421-427.
6. Federici C, Drake KM, Rigelsky CM, McNelly LN, Meade SL, Comhair SA, Erzurum SC and Aldred MA. Increased Mutagen Sensitivity and DNA Damage in Pulmonary Arterial Hypertension. *Am J Respir Crit Care Med*. 2015;192:219-28.
7. Spiekermann S, Schenk K and Hoepfer MM. Increased xanthine oxidase activity in idiopathic pulmonary arterial hypertension. *Eur Respir J*. 2009;34:276.

8. Bowers R, Cool C, Murphy RC, Tudor RM, Hopken MW, Flores SC and Voelkel NF. Oxidative stress in severe pulmonary hypertension. *Am J Respir Crit Care Med*. 2004;169:764-9.
9. Grobe AC, Wells SM, Benavidez E, Oishi P, Azakie A, Fineman JR and Black SM. Increased oxidative stress in lambs with increased pulmonary blood flow and pulmonary hypertension: role of NADPH oxidase and endothelial NO synthase. *Am J Physiol Lung Cell Mol Physiol*. 2006;290:L1069-77.
10. Redout EM, van der Toorn A, Zuidwijk MJ, van de Kolk CW, van Echteld CJ, Musters RJ, van Hardeveld C, Paulus WJ and Simonides WS. Antioxidant treatment attenuates pulmonary arterial hypertension-induced heart failure. *Am J Physiol Heart Circ Physiol*. 2010;298:H1038-47.
11. Redout EM, Wagner MJ, Zuidwijk MJ, Boer C, Musters RJ, van Hardeveld C, Paulus WJ and Simonides WS. Right-ventricular failure is associated with increased mitochondrial complex II activity and production of reactive oxygen species. *Cardiovasc Res*. 2007;75:770-81.
12. Abe K, Toba M, Alzoubi A, Ito M, Fagan KA, Cool CD, Voelkel NF, McMurtry IF and Oka M. Formation of plexiform lesions in experimental severe pulmonary arterial hypertension. *Circulation*. 2010;121:2747-54.
13. Jernigan NL, Naik JS, Weise-Cross L, Detweiler ND, Herbert LM, Yellowhair TR and Resta TC. Contribution of reactive oxygen species to the pathogenesis of pulmonary arterial hypertension. *PLoS One*. 2017;12:e0180455.
14. Stenmark KR, Frid MG, Graham BB and Tudor RM. Dynamic and Diverse Changes in the Functional Properties of Vascular Smooth Muscle Cells in Pulmonary Hypertension. *Cardiovasc Res*. 2018.
15. Tudor RM, Groves B, Badesch DB and Voelkel NF. Exuberant endothelial cell growth and elements of inflammation are present in plexiform lesions of pulmonary hypertension. *The American journal of pathology*. 1994;144:275-85.
16. Majka SM, Skokan M, Wheeler L, Harral J, Gladson S, Burnham E, Loyd JE, Stenmark KR, Varella-Garcia M and West J. Evidence for cell fusion is absent in vascular lesions associated with pulmonary arterial hypertension. *Am J Physiol Lung Cell Mol Physiol*. 2008;295:L1028-39.
17. Meloche J, Pflieger A, Vaillancourt M, Paulin R, Potus F, Zervopoulos S, Graydon C, Courboulin A, Breuils-Bonnet S, Tremblay E, Couture C, Michelakis ED, Provencher S and Bonnet S. Role for DNA Damage Signaling in Pulmonary Arterial Hypertension. *Circulation*. 2013.
18. Olcina M, Lecane PS and Hammond EM. Targeting hypoxic cells through the DNA damage response. *Clinical cancer research : an official journal of the American Association for Cancer Research*. 2010;16:5624-9.
19. Pandey RN, Wang TS, Tadjuidje E, McDonald MG, Rettie AE and Hegde RS. Structure-Activity Relationships of Benzbromarone Metabolites and Derivatives as EYA Inhibitory Anti-Angiogenic Agents. *PLoS One*. 2013;8:e84582.
20. Tadjuidje E, Wang TS, Pandey RN, Sumanas S, Lang RA and Hegde RS. The EYA Tyrosine Phosphatase Activity Is Pro-Angiogenic and Is Inhibited by Benzbromarone. *PLoS One*. 2012;7:e34806.
21. Wang Y, Pandey RN, Riffle S, Chintala H, Wikenheiser-Brokamp KA and Hegde RS. The Protein Tyrosine Phosphatase Activity of Eyes Absent Contributes to Tumor Angiogenesis and Tumor Growth. *Mol Cancer Ther*. 2018;17:1659-1669.
22. Wang Y, Tadjuidje E, Pandey RN, Stefater JA, 3rd, Smith LE, Lang RA and Hegde RS. The Eyes Absent Proteins in Developmental and Pathological Angiogenesis. *The American journal of pathology*. 2016;186:568-78.

Reviewers' comments:

Reviewer #1 (Remarks to the Author):

In the revised version of the study examining the role of Eyes Absent3 (EYA3) as a target in experimental pulmonary hypertension, the authors have provided additional studies to evaluate EYA3 in pulmonary artery endothelial cells. As mentioned in prior review, the figures continue to show that much of the DNA damage present in the models studied is present outside of pulmonary artery smooth muscle cells indicating that other mechanisms are in play. The suggestion that inhibition of EYA3 prevents and reverses established disease solely by effects on the smooth muscle cells is therefore not supported fully by the data presented. To prove that smooth muscle EYA3 is responsible for the pathophenotype, a smooth muscle cell specific knockout of EYA3 was needed. This is especially important as intimal thickening with EC proliferation and EMT also occurs in the disease. This overall issue as well as the translatability to clinic are never quite resolved.

1. The inability to provide further hemodynamic information necessitates that the authors are more cautious about interpretation of their results as they can't distinguish between reverse remodeling attributable to direct effects on the pulmonary vasculature versus secondary effects due to improved systemic hemodynamics. Although this may seem like a minor point, this is the criteria used clinically to determine if patients are medication responders.
2. Densitometry numbers for endothelial cell expression of EYA3 in Fig. 1b are confusing. If expression in controls is normalized for loading, then the numbers aren't correct. If anything, the blot suggests that EYA3 is decreased in endothelial cells. Since there is evidence of endothelial DNA damage in vessel endothelial cells (Fig. 7), it should be acknowledged that it is likely that other DNA repair effectors or mechanisms are involved in disease progression.
3. While H₂O₂ is a stressor for the endothelium, it is not chronic hypoxia and doesn't recapitulate the entire compendium of molecular changes that occur with this challenge. If you want to make the argument that chronic hypoxia or oxidant stress leads to the phenotype, then the cells should be stressed with the appropriate challenge.
4. Similarly, comments that DNA damage in vivo result from hypoxia or oxidant stress are unfounded as these metrics were never measured and the comments are speculative. This should be measured or removed and speculation reserved for the Discussion.
5. Fig. 1 IHC of EYA3 – EYA3 expression is not clear from the images, even with the red arrows. If this is brown over blue nuclei then difficult to see in images provided. By the methodology used it isn't clear if the reader is mean to observe nuclear or cytoplasmic EYA3.
6. The vessels shown in Fig. 6 and Fig. 8 from mouse lung sections to show remodeled pulmonary vessels are concerning. At low magnification the vessels appear "remodeled;" however, when magnified the lumen is not occluded as a result of cellular proliferation but this appears to be

proteinaceous material with no apparent nuclei. This is seen only in the Sugden-treated mice (controls or EyaD262N) but not the controls. Either there is a problem with processing of tissue in both Sugden-treated groups or a problem with the model.

7. To prove conclusively that smooth muscle EYA3 is responsible for the pathophenotype, a smooth muscle cell specific knockout of EYA3 is needed. This should be addressed as a limitation.

8. The comment in the Discussion that pulmonary vascular remodeling in PAH is reversible is, in fact, completely wrong and misquotes the paper cited. This needs to be changed.

Reviewer #2 (Remarks to the Author):

The authors have addressed all my previous concerns sufficiently.

Reviewer #3 (Remarks to the Author):

The authors have addressed most of my concerns. I support the publication of this manuscript. However, I would like the authors not to definitely claim EYA3's role in DNA damage as the only mechanism. EYA3 could have other substrates that impact cell survival. The authors provided some correlative evidence for decreased DNA repair, however, it is not proved that the observed is really through Y142 phosphorylation or not.

Thank-you once again for very detailed reviews that have helped us clarify key aspects of this manuscript. Specific reviewer comments are listed below along with our responses (in *blue italics*).

Reviewer #1 (Remarks to the Author):

In the revised version of the study examining the role of Eyes Absent3 (EYA3) as a target in experimental pulmonary hypertension, the authors have provided additional studies to evaluate EYA3 in pulmonary artery endothelial cells. As mentioned in prior review, the figures continue to show that much of the DNA damage present in the models studied is present outside of pulmonary artery smooth muscle cells indicating that other mechanisms are in play. The suggestion that inhibition of EYA3 prevents and reverses established disease solely by effects on the smooth muscle cells is therefore not supported fully by the data presented. To prove that smooth muscle EYA3 is responsible for the pathophenotype, a smooth muscle cell specific knockout of EYA3 was needed. This is especially important as intimal thickening with EC proliferation and EMT also occurs in the disease. This overall issue as well as the translatability to clinic are never quite resolved.

While our data suggests that EYA3 is elevated in SMC from PAH patients, the effects of loss of EYA3 PTP activity in the studies reported here (either through pharmacological inhibition or genetic manipulation) are not smooth muscle specific. Accordingly, we do not conclude that smooth muscle cell EYA3 is the sole contributor to the observed vascular remodeling and we have been careful to avoid implying this in the manuscript. On page 11 we explicitly state: "Since the experimental strategies used here are not cell-type specific the cell-autonomous contributions of EYA3 to vascular remodeling remain to be established. "

1. The inability to provide further hemodynamic information necessitates that the authors are more cautious about interpretation of their results as they can't distinguish between reverse remodeling attributable to direct effects on the pulmonary vasculature versus secondary effects due to improved systemic hemodynamics. Although this may seem like a minor point, this is the criteria used clinically to determine if patients are medication responders.

We fully recognize the importance of functional hemodynamics, and this is acknowledged on page 12: "More extensive evaluation of functional parameters including measurement of cardiac output, pulmonary vascular resistance and RV function remains necessary."

In this report we link, for the first time, the EYA3-PTP activity with vascular remodeling. We have emphasized the limitations of the present study in the final paragraph covering the issues of functional hemodynamics, cell-autonomous contributions of EYA3 (point 7, reviewer 1) and non-DNA damage-repair mechanisms that could contribute to remodeling (reviewer 3):

"In conclusion, our studies support the EYA3-PTP pathway as a contributor to pulmonary vascular remodeling (target-validation) and demonstrate the utility of Benzarone as a tool compound for the development of EYA-PTP inhibitors as effective PAH therapeutics. Clinical translation of these findings would require the completion of critical next steps. A more comprehensive assessment of functional hemodynamics is necessary to evaluate whether the observed reverse remodeling is due to a primary effect on the pulmonary vasculature or is secondary to improved systemic hemodynamics. The cell-autonomous contributions of smooth muscle and endothelial cell EYA3, and other EYA3-modulated molecular mechanisms (in addition to DNA damage repair) that contribute to vascular remodeling need to be defined. These are the subject of ongoing studies."

2. Densitometry numbers for endothelial cell expression of EYA3 in Fig. 1b are confusing. If expression in controls is normalized for loading, then the numbers aren't correct. If anything, the blot suggests that

EYA3 is decreased in endothelial cells. Since there is evidence of endothelial DNA damage in vessel endothelial cells (Fig. 7), it should be acknowledged that it is likely that other DNA repair effectors or mechanisms are involved in disease progression.

The confusion regarding the densitometry arises from the fact that in both 1a and 1b the densitometry reflects the average of multiple experiments, while one representative blot is shown. The Western blots in Figure 1 have been edited so that the levels shown in 1a and 1b now come only from the blot shown here. The legend has been edited to reflect this change. We state in the legend and in the text that levels of EYA3 are not elevated in PAH-PAEC. We do not have convincing evidence, given the small differences observed, that levels are decreased in PAH-PAEC relative to normal PAEC.

Page 10 also states that “Some residual γ -H2AX and 53BP1 staining was present in endothelial cells after BZ treatment (Fig. a, d) possibly indicating ongoing DNA-damage and repair in surviving PAECs through non-EYA3-dependent mechanisms.”

We also extensively discuss non-EYA3 dependent DNA damage repair mechanism on page 11: “The consequences of DNA damage in pulmonary vascular cells appear to be cell-type and context-specific. In PAH-PAEC DNA damage is accompanied by increased genomic instability, apoptosis in the early stages of disease, mesenchymal transformation, and release of pro-inflammatory stimuli. PAH-PASMC on the other hand resist apoptosis and can proliferate despite the hostile microenvironment. Several DDR pathways are elevated in PASMC from PAH patients including the DNA damage repair factor poly(ADP-ribose) polymerase-1 (PARP-1), and the transcription factor FOXM1 that can promote the expression of the DNA repair protein NBS1 (Nijmegen breakage syndrome. Indeed, PARP-1 inhibition can reverse PAH in vivo and the PARP-1 inhibitor Olaparib is in clinical trials as a PAH therapeutic. “

3. While H₂O₂ is a stressor for the endothelium, it is not chronic hypoxia and doesn't recapitulate the entire compendium of molecular changes that occur with this challenge. If you want to make the argument that chronic hypoxia or oxidant stress leads to the phenotype, then the cells should be stressed with the appropriate challenge.

&

4. Similarly, comments that DNA damage in vivo result from hypoxia or oxidant stress are unfounded as these metrics were never measured and the comments are speculative. This should be measured or removed and speculation reserved for the Discussion.

H₂O₂ is used here as an experimental tool to induce DNA damage, much like hydroxyurea¹ or etoposide² is used in other studies. The fact that DNA damage is induced by H₂O₂ is documented in Supplementary Figure 2j. The relevant section on page 6 has been re-named to clarify this. The manuscript has been carefully reviewed to avoid the implication that oxidative stress causes DNA damage in vivo in our studies.

5. Fig. 1 IHC of EYA3 – EYA3 expression is not clear from the images, even with the red arrows. If this is brown over blue nuclei then difficult to see in images provided. By the methodology used it isn't clear if the reader is meant to observe nuclear or cytoplasmic EYA3.

Fig. 1 has been modified to show enlarged images where increased levels of nuclear EYA3 are present in the PH samples.

6. The vessels shown in Fig. 6 and Fig. 8 from mouse lung sections to show remodeled pulmonary vessels are concerning. At low magnification the vessels appear “remodeled;” however, when magnified the

lumen is not occluded as a result of cellular proliferation but this appears to be proteinaceous material with no apparent nuclei. This is seen only in the Sugden-treated mice (controls or EyaD262N) but not the controls. Either there is a problem with processing of tissue in both Sugden-treated groups or a problem with the model.

The intention of Figure 6 is to show muscularization, and in Fig. 8 we show DNA damage markers. H&E panels were included at the request of previous Reviewer 3. It appears from the comments here that those panels might lead to the mistaken impression that we are reporting occlusion of vessels in the mouse model. As has been extensively reported, the mouse Su-Hx vascular pathology is characterized by medial hypertrophy, but no obstructed vessels. The H&E images have been removed to avoid any suggestion that occluded lumens are present and the lack of occlusion is specifically referred to on page 9:

“Mouse models of PH have limitations; while increased RVSP has been reported in numerous studies immediately after 3 weeks in hypoxia, there is no evidence of significant or sustained angio-obliterative disease in mice. Likewise, we did not observe either angio-obliteration or occluded lumens. Despite this limitation, the Su-Hx mouse model allows the use of transgenic mice and are an informative adjunct to the pharmacological approach used in the rat model.”

7. To prove conclusively that smooth muscle EYA3 is responsible for the pathophenotype, a smooth muscle cell specific knockout of EYA3 is needed. This should be addressed as a limitation.

We have done so in the final paragraph of the discussion.

8. The comment in the Discussion that pulmonary vascular remodeling in PAH is reversible is, in fact, completely wrong and misquotes the paper cited. This needs to be changed.

This section has been re-worded and references appropriately located to clarify what we meant. “Muscularization of vascular walls and phenotypic alteration of endothelial cells contribute to vascular remodeling, the predominant pathological characteristic of PAH. The occlusion of small arterioles caused by the proliferation of phenotypically altered and transdifferentiated PAECs is considered irreversible. But there has been some suggestion that PASMOC can shift between proliferative and non-proliferative states raising the possibility that this aspect of vascular remodeling may be reversible. The data presented here show that either pharmacological inhibition or genetic loss of the EYA3 protein tyrosine phosphatase can attenuate the development of vascular remodeling and substantially reverse established PH in the rat Sugden-hypoxia model. The predominant consequence of loss of EYA3 PTP activity reported here is reduced muscularization of small pulmonary arteries. “

Reviewer #2 (Remarks to the Author):

The authors have addressed all my previous concerns sufficiently.

Reviewer #3 (Remarks to the Author):

The authors have addressed most of my concerns. I support the publication of this manuscript. However, I would like the authors not to definitely claim EYA3's role in DNA damage as the only mechanism. EYA3 could have other substrates that impact cell survival. The authors provided some correlative evidence for decreased DNA repair, however, it is not proved that the observed is really through Y142 phosphorylation or not.

Other mechanisms that could contribute to the observed effects are discussed in the conclusions section and re-iterated in the concluding sentence “The cell-autonomous contributions of smooth muscle and

endothelial cell EYA3, and other EYA3-modulated molecular mechanisms (in addition to DNA damage repair) that contribute to vascular remodeling need to be defined. These are the subject of ongoing studies.”

References

1. Li CG, Mahon C, Sweeney NM, Verschueren E, Kantamani V, Li D, Hennigs JK, Marciano DP, Diebold I, Abu-Halawa O, Elliott M, Sa S, Guo F, Wang L, Cao A, Guignabert C, Sollier J, Nickel NP, Kaschwich M, Cimprich KA and Rabinovitch M. PPARgamma Interaction with UBR5/ATMIN Promotes DNA Repair to Maintain Endothelial Homeostasis. *Cell Rep.* 2019;26:1333-1343 e7.
2. Meloche J, Pflieger A, Vaillancourt M, Paulin R, Potus F, Zervopoulos S, Graydon C, Courboulin A, Breuils-Bonnet S, Tremblay E, Couture C, Michelakis ED, Provencher S and Bonnet S. Role for DNA damage signaling in pulmonary arterial hypertension. *Circulation.* 2014;129:786-97.

REVIEWERS' COMMENTS:

Reviewer #1 (Remarks to the Author):

The authors have responded to prior comments and clarified a number of issues through revisions tot the text.

Reviewer #1 (Remarks to the Author):

The authors have responded to prior comments and clarified a number of issues through revisions tot the text.

No changes have been requested by the reviewers.